# COBRA: Contextual Bandit Algorithm for Ensuring Truthful Strategic Agents

## Abstract

This paper considers a contextual bandit problem involving multiple agents, where a learner sequentially observes the contexts and the agents' reported arms, and then selects the arm that maximizes the system's overall reward. Existing work in contextual bandits assumes that agents always truthfully report their arms, which is unrealistic in many real-life applications. For instance, consider an online platform with multiple sellers; some sellers may misrepresent product features to gain an advantage, such as having the platform preferentially recommend their products to its users. To address this challenge, we propose an algorithm, COBRA, for contextual bandit problems involving strategic agents that disincentivize their strategic behavior without using any monetary incentives, while having incentive compatibility and a sub-linear regret guarantee. Our experimental results also validate our theoretical results and the different performance aspects of COBRA.

## 1 Introduction

Contextual bandit (Slivkins, 2019; Lattimore and Szepesvári, 2020) is a sequential decision-making framework in which a learner selects an arm for a given context to maximize its total reward. Unlike traditional multi-armed bandits (Auer et al., 2002; Garivier and Cappé, 2011; Agrawal and Goyal, 2012), contextual bandit algorithms use additional information, such as user profile, location, and purchase history, to make more informed and personalized decisions (Li et al., 2010). Contextual bandits have many real-life applications in personalized decision-making, such as online recommendation systems (Slivkins, 2019), online advertising (Lattimore and Szepesvári, 2020), and clinical trials (Chow and Chang, 2006; Aziz et al., 2021), where the best recommendation depends on the context.

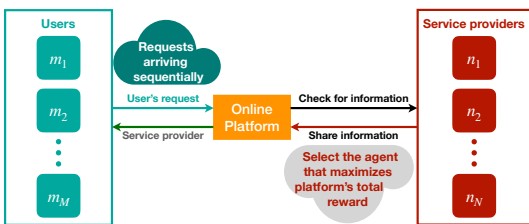

Figure 1: Example of a contextual bandit problem with strategic agents: Consider an online platform recommending service providers (agents) to users (context) who arrive sequentially. Since service providers can misreport their private information to receive favorable recommendations, the platform must implement a mechanism incentivizing truthful reporting. With accurate private information, the platform can recommend the best service provider, thereby improving the overall user experience.

Many real-life applications of contextual bandits involve multiple strategic agents, from which the learner must select one to recommend based on the given context. As illustrated in Fig. 1, consider an online platform with multiple service providers (agents), where the platform must recommend one provider to a user (context). In such settings, service providers can strategically misreport their information to influence the platform's decisions and increase their utilities by increasing their chances of being recommended (Resnick and Sami, 2007; Zhang et al., 2019). For example, an online food delivery platform wants to maximize the overall user experience by selecting and presenting the best restaurant options when a user searches for a specific type of food. Since users tend to order from restaurants listed at the top of search results (Malaga, 2008), restaurants are incentivized to misrepresent their menu offerings to appear more prominently for specific food categories. This misreporting creates a challenge: If users consistently encounter misleading restaurant listings that do not match their preferences, their experience with the platform will worsen; in the worst case, they may switch to competing platforms. Similar examples also include

personalized pricing, where agents manipulate features to influence service prices (Liu et al., 2024); algorithmic trading, where arms correspond to trading strategies, and rewards depend on evolving market conditions influenced by external factors such as Twitter feeds, secondary market behavior, and local trends, which can be misreported (Zeng et al., 2024a); and firms allocating budget-constrained computing resources to self-interested research teams that might misreport their demand to secure larger allocations, especially around conference deadlines (Zeng et al., 2024b).

These real-life applications highlight the importance of designing contextual bandit algorithms that discourage strategic misreporting by agents. However, most existing contextual bandit algorithms overlook the strategic behavior of agents, which can result in suboptimal agent selection. We bridge this gap by designing a contextual bandit algorithm that accounts for potential misreporting and ensures that reporting arm features truthfully is the best (*dominant*) strategy for agents. Specifically, this paper answers the following question: ***How to design an efficient incentive-compatible contextual bandit algorithm for settings where strategic agents may misreport their true features?***

To address this question, we propose a contextual bandit algorithm, COBRA, that discourages strategic misreporting without relying on any monetary incentives, while having incentive compatibility and sub-linear regret guarantees under some mild assumptions. Designing incentive-compatible contextual bandit algorithms presents following key challenges, which we address using novel techniques.

① **Detecting strategic misreporting.** Existing contextual bandit algorithms (Slivkins, 2019; Lattimore and Szepesvári, 2020) typically assume that agents truthfully report the features of their arms, which may not hold in many practical applications to gain an advantage. Thus, a key challenge is to reliably identify whether an agent is strategically misreporting arm features to manipulate outcomes. To overcome this challenge, we introduce a novel method, the *Leave-One-Out-based Mechanism (LOOM) for identifying misreporting agents*, which draws inspiration from the Vickrey-Clarke-Groves (VCG) mechanism (Vickrey, 1961; Clarke, 1971; Groves, 1973) and uses the reported arm features of other agents to identify the misreporting agent within the contextual bandit setting.

② **Arm-selection under strategic misreporting.** Strategic misreporting of arm features introduces bias into the reward estimator, breaking the theoretical guarantees of existing contextual bandit algorithm (Kleine Buening et al., 2024). To address this bias in the reward estimator, we first introduce the notion of an LOOM-compatible contextual bandit algorithm. An LOOM-compatible contextual bandit algorithm (e.g., algorithms based on upper confidence bound (Abbasi-Yadkori et al., 2011; Chowdhury and Gopalan, 2017)) can integrate *LOOM as a post-processing step after each round to identify the misreporting agent*. This integration ensures that the arm-selection strategy of the underlying LOOM-compatible contextual bandit algorithm incurs no performance loss when all agents report truthfully, while adaptively correcting for bias once LOOM identifies a strategic agent.

③ **Bounding regret due to strategic misreporting.** The arm-selection strategy relies on a reward estimator that may be biased by using strategically misreported arm features. This bias can lead to the selection of suboptimal arms, resulting in additional loss from not choosing the optimal arms (i.e., increased regret). To quantify this additional regret, we derive high-probability upper bounds on the estimation error of the reward estimators used by LOOM. These bounds are then used to characterize the regret incurred by the underlying contextual bandit algorithm when combined with LOOM.

Building on the challenges outlined above and the novel methods we use to tackle them, we summarize the key contributions of this paper as follows:

- **Incentive-compatible mechanism.** In Section 3, we introduce LOOM, a novel mechanism for contextual bandits with strategic agents inspired by the VCG framework. Unlike VCG, LOOM discourages strategic misreporting without monetary incentives, and we show that truthful agents are, with high probability, not misidentified as a misreporting agent (see Theorem 1).

- **Incentive-compatible contextual bandit algorithm.** We propose COBRA in Section 4 that uses LOOM to disincentivize agents from misreporting. We prove that COBRA achieves $\tilde{O}(d\sqrt{T})$-NE (i.e., truthfulness leads to an approximate Nash equilibrium) and regret $\tilde{O}(d\sqrt{T})$ when agents report truthfully (see Theorem 2 for linear and Theorem 3 for non-linear reward function). Under some mild assumptions, we prove that COBRA has regret at most $\tilde{O}(d\sqrt{T} + \sqrt{NT})$ under every Nash equilibrium (see Theorem 4 for linear and Theorem 5 for non-linear reward function), where $N$ is the number of agents, $d$ is the dimension of the context vector, and $T$ is the number of contexts.

- **Empirical results.** In Section 5, our experimental results on contextual bandit instances with strategic agents corroborate our theoretical results and validate the performance of COBRA.

## 1.1 Related Work

This section focuses on the most relevant work to our setting, i.e., strategic multi-armed and contextual bandits. We discuss related topics, such as contextual bandits and strategic learning, in Section A.1.

**Strategic multi-armed bandits.** To the best of our knowledge, Braverman et al. (2019) first studied a strategic variant of the multi-armed bandit problem, considering a scenario in which the selected arm shares a fraction of its reward with the learner. Within this setting, they designed an incentive-compatible mechanism. More recently, Yahmed et al. (2024) further built upon Braverman et al. (2019), proposing an algorithm that rewards arms based on their reported values. Their algorithm also enjoys desirable properties such as incentive compatibility and sub-linear regret. Additionally, Yin et al. (2022) studied an online allocation problem that maximizes social welfare under fairness constraints in a strategic setting. They assume that valuations are unknown to the algorithm but follow an independent and identical distribution (IID). Their results show that when agents truthfully reveal their information, the mechanism maximizes social welfare while also achieving a sub-linear regret guarantee compared to the offline optimal policy. Our mechanism design follows a similar spirit but is applied to a different problem setting. Moreover, Feng et al. (2020) and Dong et al. (2022) explore the robustness of bandit learning against strategic manipulation, assuming a bounded manipulation budget. Esmaeili et al. (2023); Shin et al. (2022) investigate multi-armed bandits with replicas, where strategic agents can submit multiple copies of the same arm. Kleine Buening et al. (2023) integrate multi-armed bandits with mechanism design for online recommendations.

**Strategic contextual bandits.** Our work is closely related to Kleine Buening et al. (2024), which considers the strategic agents in a linear contextual bandit framework. Their method uses past allocation history to design agent-specific estimators that detect misreports with high probability, which may not be practical, particularly when the true reward function is unknown, and there is no external baseline for comparison. In contrast, our method is inspired by the VCG mechanism (Vickrey, 1961; Clarke, 1971; Groves, 1973), using the reported arm features of other agents to identify misreports and supports non-linear reward function. Recent work by Hu and Duan (2025) introduce a Bayesian contextual linear bandit framework in a similar spirit, with non-repeated agent interactions, employing a linear programming-based approach to design an incentive-compatible mechanism. However, our setting significantly differs due to inherent repeated interactions in contextual bandits.

## 2 Contextual Bandits with Strategic Agents

**Contextual bandits.** This paper studies a contextual bandit problem with strategic agents who aim to maximize their utility (i.e., number of pulls) by strategically misreporting their arm's feature to the learner, while the learner's goal is to select the agent for a given context that maximizes the total reward. Our problem setting differs from standard contextual bandits as the arm features can be strategically manipulated by the agents to maximize their own utility. Let $\mathcal{C}$ be the set of all contexts and $\mathcal{A}$ be the set of all arms of all agents. Let $\mathcal{N}$ be the set of all agents and $N_t \leq |\mathcal{N}|$ denote the number of active agents at time $t$. For brevity, we use $\mathcal{X} \subset \mathbb{R}^d$ to denote the set of all context-arm feature vectors, and $x_{t,a} = \varphi(c_t, a) \in \mathcal{X}$ to represent the feature vector associated for context $c_t$ and arm $a \in \mathcal{A}$, where $\varphi : \mathcal{C} \times \mathcal{A} \to \mathcal{X}$ is a feature map and $\|x\|_2^2 \leq L, \ \forall x \in \mathcal{X}$. At the start of round $t$, the environment generates a context $c_t \in \mathcal{C}$ and each agent $n \in \mathcal{N}_t \subseteq \mathcal{N}$ reports arm features, denoted by $a_t^{(n)} \in \mathcal{A}_t \subset \mathcal{A}$, where $\mathcal{N}_t$ is the set of active agents in round $t$ and $\mathcal{A}_t = \{a_t^{(n)}\}_{n \in \mathcal{N}_t}$. The learner then selects an arm $a_t \in \mathcal{A}_t$ to recommend and observes a stochastic reward, denoted by $y_t \doteq f(x_{t,a_t}) + \varepsilon_t$, where $y_t \in \mathbb{R}$, $f : \mathcal{X} \to \mathbb{R}$ is an unknown reward function, and $\varepsilon_t$ is a zero-mean $R$-sub-Gaussian noise. For simplicity, we assume that the agent only reports one arm in each round so that we can use 'agent' and 'arm' interchangeably in the paper.[1]

**Strategic manipulations by agents.** A strategic agent can misreport the features of their arm by manipulating them such that the agent is selected more often, thereby maximizing its utility. Let $x_{t,a}^\star$ be the true arm feature vector and $x_{t,a}$ be the reported arm feature vector for context $c_t$ and arm $a$. Although an agent can strategically manipulate the arm feature vector, we assume the observed reward

---

[1]Our framework are more general and can also apply to settings where agents can report multiple arms per round, e.g., sellers offering multiple variants of the same product on an online platform. We also want to highlight that all agent-related computations can be performed in parallel, as they are independent of one another.

only depends on the true arm feature vector.[2] To maximize the total reward, our aim is to design a contextual bandit algorithm incorporating an incentive-compatible mechanism that ensures truthful reporting (i.e., $x_{t,a} = x_{t,a}^{\star}$, $\forall t \geq 1, a \in \mathcal{A}$) is the dominant strategy for all agents.

**Incentive-Compatible algorithm.** Let $\sigma_n$ denote the strategy of agent $n \in \mathcal{N}$, which is history-dependent and maps the true features of their arms to reported features. We use $\boldsymbol{\sigma}_{-n}$ to denote the strategies of all agents other than agent $n$, and $\boldsymbol{\sigma} = (\sigma_1, \sigma_2, \ldots, \sigma_N)$ to represent the full strategy profile of all agents. We first define what it means for an agent to be truthful.

**Definition 1** (Truthful). *An agent $n \in \mathcal{N}$ is said to be truthful if agent reports the true features of their arms to the learner in each round, i.e., $x_{t,a} = x_{t,a}^{\star}$ for all $t \geq 1$ and $a$ denotes agent's arm.*

We use $\sigma_n^*$ to denote the truthful strategy for the agent $n$ and $\boldsymbol{\sigma}^* = (\sigma_1^*, \sigma_2^*, \ldots, \sigma_N^*)$ to represent the vector of the truthful strategy for all agents. Next, we formally define the utility of an agent $n$ in our setting. Let $S_T(n) \doteq \sum_{t=1}^{T} \mathbb{1}(\text{arm } a_t \text{ belongs to agent } n)$ denote the number of times agent $n$ is selected by the learner up to round $T$. Each agent's objective is to maximize the expected number of $S_T(n)$. Therefore, the utility of agent $n$ is given by $u_a(\boldsymbol{\sigma}) \doteq \mathbb{E}\left[ S_T(a) \mid \boldsymbol{\sigma} \right]$, where we conditioned on all agents strategies $\boldsymbol{\sigma}$. In the following, we define the notion of $\varepsilon$-Nash equilibrium (NE), in which no agent has more than $\varepsilon$ incentive to deviate from the truthful reporting strategy.

**Definition 2** ($\varepsilon$-Nash Equilibrium). *Let $\varepsilon > 0$ and $T > 0$. We say that $\boldsymbol{\sigma} = (\sigma_1, \sigma_2, \ldots, \sigma_N)$ forms a $\varepsilon$-Nash equilibrium if any deviating strategy $\sigma_a'(\neq \sigma_a)$ for any agent $a \in \mathcal{A}$, the following holds:*
$$\mathbb{E}\left[ S_T(a) \mid \sigma_a, \boldsymbol{\sigma}_{-a} \right] \geq \mathbb{E}\left[ S_T(a) \mid \sigma_a', \boldsymbol{\sigma}_{-a} \right] - \varepsilon.$$

We next define incentive compatibility for a contextual bandit algorithm in terms of Nash equilibrium.

**Definition 3** (Incentive Compatible). *A contextual bandit algorithm is incentive compatible if truthfulness is a Nash equilibrium, i.e., reporting the true arm features maximizes each agent's utility.*

**Performance measure.** Let $a_t^{\star}$ denote the optimal arm (agent) for context $c_t$ having the maximum expected reward, i.e., $a_t^{\star} = \operatorname{argmax}_{a \in \mathcal{A}_t} f(x_{t,a})$. After selecting arm $a_t$, the learner incurs a penalty $r_t$, where $r_t = f(x_{t,a_t^{\star}}^{\star}) - f(x_{t,a_t}^{\star})$. Our aim is to learn a sequential policy that selects an arm for a given context such that the learner's total penalty for not selecting the optimal arm (or *cumulative regret*) is as minimal as possible. However, the performance of the contextual bandit algorithm depends on the incentive-compatible mechanism for the strategic agents whose strategy profile is represented by $\boldsymbol{\sigma} = (\sigma_1, \ldots, \sigma_N)$. We use strategic regret as a performance measure of a sequential policy $\pi$ for which the agents act according to a Nash equilibrium under policy $\pi$. Specifically, for $T$ rounds and $\boldsymbol{\sigma} \in \operatorname{NE}(\pi)$, the strategic regret of a policy $\pi$ that selects arm $a_t$ in the round $t$ is

$$\mathfrak{R}_T(\pi, \boldsymbol{\sigma}) \doteq \sum_{t=1}^{T} \left( f(x_{t,a_t^{\star}}^{\star}) - f(x_{t,a_t}^{\star}) \right). \tag{1}$$

A policy $\pi$ is a good policy if it has sub-linear regret, i.e., $\lim_{T \to \infty} \mathfrak{R}_T(\pi, \boldsymbol{\sigma})/T = 0$. This implies that, as $T$ increases, the policy $\pi$ will eventually start selecting optimal arms for the given contexts.

## 3 Leave-One-Out-based Mechanism (LOOM)

In the contextual bandit setting, designing an incentive-compatible mechanism that ensures truthful reporting of arm features by agents is challenging due to limited access to true contexts, the potential for strategic misreporting, noisy reward feedback, and unknown reward function parameters. These challenges naturally raise the question: *How can we design a mechanism that effectively incentivizes strategic agents to report truthfully?* To overcome this, we propose a method, *Leave-One-Out-based Mechanism (LOOM) for identifying misreporting agents*, which is inspired by the Vickrey-Clarke-Groves (VCG) framework (Vickrey, 1961; Clarke, 1971; Groves, 1973) and uses the reported arm features of other agents to identify misreporting agent. To identify whether an agent $a$ is misreporting (i.e., *over-reporting* arm features to increase its expected reward, such that $f(x_{t,a}) > f(x_{t,a}^{\star})$), LOOM uses three key components: ① a pessimistic estimate of the agent's total expected reward, derived from past data of all other agents, ② an optimistic estimate of the agent

---

[2]Sellers can misrepresent product features on the e-commerce platform such that it becomes a top recommendation. However, it cannot change the actual physical quality and nature of the product.

total reward, based on the observed rewards when agent $a$ is selected, ③ a statistical test that uses these estimates to identify if agent $a$ is over-reporting with high probability.

① **Pessimistic estimate of the agent's total expected reward.** Since the true reward function is unknown, LOOM estimates it using past observations (context-arm features and rewards) from all agents except agent $a$, ensuring the estimator is not influenced by agent $a$. Next, we formally introduce the notion of a *LOOM-compatible* contextual bandit algorithm, which refers to a class of algorithms that can integrate LOOM as a post-processing step after each round to identify the over-reporting agent. Let $\mathcal{O}_t$ denote the past observations from all agents at the beginning of round $t$ and $\mathcal{O}_{t,-a}$ represent the past observations from all agents except agent $a$. Let $f_t$ and $f_{t,-a}$ represent the estimate of reward function $f$ using observations $\mathcal{O}_t$ and $\mathcal{O}_{t,-a}$, respectively, at the end of round $t$.

**Definition 4** (**LOOM-Compatible Contextual Bandit Algorithm**). *Any contextual bandit algorithm $\mathfrak{A}$ is LOOM-compatible if the following holds: (i) The estimated function $f_t^{\mathfrak{A}}$ from $\mathcal{O}_t$, with probability $1 - \delta$, satisfies: For any $x \in \mathcal{X}$: $|f_t^{\mathfrak{A}}(x) - f(x)| \leq h(x, \mathcal{O}_t)$. and (ii) The estimated function $f_{t,-a}^{\mathfrak{A}}$ from $\mathcal{O}_{t,-a}$, with probability $1 - \delta$, satisfies: For any $x \in \mathcal{X}$: $|f_{t,-a}^{\mathfrak{A}}(x) - f(x)| \leq h(x, \mathcal{O}_{t,-a})$. Here, the value of $h(x, \cdot)$ depends on $x$, past observations ($\mathcal{O}_t$ or $\mathcal{O}_{t,-a}$), and $\mathfrak{A}$.*

Many contextual bandit algorithms like Lin-UCB (Chu et al., 2011) (as shown in Section 4), UCB-GLM (Li et al., 2017), IGP-UCB (Chowdhury and Gopalan, 2017), GP-TS (Chowdhury and Gopalan, 2017), Neural-UCB (Zhou et al., 2020), and Neural-TS (Zhang et al., 2021) are LOOM-compatible. Depending on the problem setting, any suitable LOOM-compatible contextual bandit algorithm can be used, where arms are selected according to the algorithm's inherent arm selection strategy, and LOOM is used to identify strategic agents. The value of $h(x, \mathcal{O}_t)$ and $h(x, \mathcal{O}_{t,-a})$ provide the upper bounds on the estimated rewards with respect to the true reward function. This value depends on the problem and the choice of contextual bandit algorithm $\mathfrak{A}$ and its associated hyperparameters. Note that the assumptions required by contextual bandit algorithms must also hold in our setting, as they directly influence the performance of our proposed algorithm through $h(x, \mathcal{O}_t)$ and $h(x, \mathcal{O}_{t,-a})$. See Table 1 in supplementary material in Section C for different values of $h(x, \mathcal{O}_t)$.

Henceforth, we assume that the underlying contextual bandit algorithm is LOOM-compatible and omit the superscript $\mathfrak{A}$ in estimators ($f_t^{\mathfrak{A}}$ and $f_{t,-a}^{\mathfrak{A}}$) superscript for notational simplicity. For any $x \in \mathcal{X}$, if $|f_{t,-a}(x) - f(x)| \leq h(x, \mathcal{O}_{t,-a})$ holds with probability $1 - \delta_{t,a}$, then $\text{LCB}_{t,-a}(x) = f_{t,-a}(x) - h(x, \mathcal{O}_{t,-a})$ is a pessimistic estimates of the expected reward for $x$, which also holds with probability $1 - \delta_{t,a}$ (see Lemma 6 in Section C). We further define $\text{LCB}_{t,a}^{(x)} = \sum_{s=1, a_s=a}^{t} \text{LCB}_{t,-a}(x_{s,a_s})$ to denote the pessimistic estimates of the agent $a$'s total expected reward. We assume that $\text{LCB}_{t,a}^{(x)}$ holds with probability at least $1 - \delta_{t,a}^x$ (more details are provided in Section C).

② **Optimistic estimate of the agent total reward.** Since the observed reward depends only on the true feature vector, the learner receives a noisy reward, where the noise is sub-Gaussian. Our following result provides an optimistic estimate of the agent $a$'s total expected reward.

**Lemma 1.** *Let $S_t(a)$ be the number of times that agent $a$ is selected until round $t$, and $\varepsilon_s$ be $R$-sub-Gaussian in the observed reward $y_s$, where $1 \leq s \leq t$. Then, with probability at least $1 - \delta_{t,a}^y$*

$$\sum_{s \leq t, a_s = a} f(x_{s,a_s}^{\star}) \leq \sum_{s \leq t, a_s = a} y_s + \sqrt{2R^2 S_t(a) \log(1/\delta_{t,a}^y)}.$$

**Proof outline.** This result follows from applying Hoeffding inequality to the sum of sub-Gaussian random variables. The detailed proof with other missing proofs are provided in Section B.

③ **Statistical test for finding whether agent is over-reporting.** For simplicity, consider the case where the reward function is known. In this case, we say that an agent is over-reporting if the total expected reward for reported arm features exceeds the total noiseless expected reward, i.e., $\sum_{s \leq t, a_s = a} f(x_{s,a_s}^{\star}) > \sum_{s \leq t, a_s = a} \bar{y}_s$ for any $t \geq 1$, where $\bar{y}_s$ is the noiseless expected reward. However, since the reward function is unknown and the observed reward is noisy in practice, we assess over-reporting using optimistic and pessimistic estimates of the expected rewards. We define $\text{UCB}_{t,a}^{(y)} = \sum_{s=1, a_s=a} y_s + \sqrt{2S_t(a) \log(1/\delta_{t,a}^y)}$ as the optimistic estimate of the sum of the agent $a$'s expected rewards that holds with probability at least $1 - \delta_{t,a}^y$. Therefore, an agent $a$ over-eporting the true arm features with probability at least $1 - \delta_{t,a}^x - \delta_{t,a}^y$ if the following condition holds:

$$\boxed{\textbf{LOOM Condition: } \text{LCB}_{t,a}^{(x)} > \text{UCB}_{t,a}^{(y)}.} \tag{2}$$

By eliminating the agent who satisfy Eq. (2) from future rounds, this LOOM condition incentivizes agents to report truthfully. Our next result shows that when an agent $a$ always reports truthfully, i.e., $x_{t,a} = x^\star{}_{t,a}$ for all $t \geq 1$, it does not get eliminated with high probability at least $1 - \delta^x_{t,a} - \delta^y_{t,a}$.

**Theorem 1.** *Let agent $a$ reports truthfully. Then, LOOM does not eliminate agent $a$ with high probability at least $1 - \delta^x_{t,a} - \delta^y_{t,a}$.*

**Proof outline.** The key idea of the proof is to apply the confidence ellipsoid lemma alongside high-probability upper bounds on the noisy reward and lower bounds on the expected reward for agent $a$'s reported arm features. Additional details are provided in the supplementary material.

**Impact of arm feature distribution on LOOM identification.** The performance of LOOM depends on how well the remaining agents represent the distribution of a over-eporting agent's arm feature, as captured by the term $h(x, \mathcal{O}_{t,-a})$ in $\text{LCB}_{t,-a}(x)$. If the over-eporting agent's distribution is well represented, $h(x, \mathcal{O}_{t,-a})$ is going to be small, making the over-eporting agent easier to identify; otherwise, a large $h(x, \mathcal{O}_{t,-a})$ makes identification harder. This structural property is central to our setting, as it allows estimating an agent's expected reward using the past observations from other agents under the shared reward function $f$ (an illustrative example is provided in Section D.1, along with discussion of LOOM, including its failure cases, such as heterogeneous agents, multiple strategic agents, and collusion, as well as its connections to existing work (Kleine Buening et al., 2024)).

**Remark 1** (Agent under-reporting.)**.** *Agents have no incentive to under-report, as it typically reduces their likelihood of being selected by the learner. Instead, they are more inclined to over-report to increase their chances of being selected. However, as noted in (Kleine Buening et al., 2024), there are some cases where under-reporting may yield a small gain. Our proposed method, LOOM, is specifically designed to detect over-reporting and does not capture under-reporting. Developing a mechanism that can reliably detect both under-reporting and over-reporting remains an open problem.*

## 4 INCENTIVE-COMPATIBLE CONTEXTUAL BANDIT ALGORITHM: COBRA

In this section, we present our contextual bandit algorithm, COBRA, which is specifically designed to ensure strategic agents report truthfully. To bring out our key ideas and results, we restrict our setting to linear reward functions and later extend our results to non-linear reward functions.

**Linear reward function.** We first consider the setting where the underlying reward function is linear, i.e., $f(x) = \theta_\star^\top x$ in which $\theta_\star \in \mathbb{R}^d$ is the unknown parameter. At the beginning of round $t$, the learner observes the randomly generated context $c_t \in \mathcal{C}$ and the set of reported arm features $\mathcal{A}_t$. After selecting the arm $a_t$, the learner observes stochastic reward $y_t = \theta_\star^\top x_{t,a_t} + \varepsilon_t$, where $x_{t,a_t} = \varphi(c_t, a_t)$ and $\varepsilon_t$ is $R$-sub-Gaussian. We estimate the unknown parameter $\theta_\star$ using the available observations of context-arm features and corresponding rewards at the beginning of round $t$, denoted by $\mathcal{O}_t \doteq \{(x_{s,a_s}, y_s)\}_{s=1}^{t-1}$, as follows: $\hat{\theta}_t \doteq V_t^{-1} \sum_{s=1}^{t-1} x_{s,a_s} y_s$, where $V_t \doteq \lambda I_d + \sum_{s=1}^{t-1} x_{s,a_s} x_{s,a_s}^\top$, $I_d$ is the $d \times d$ identity matrix, and $\lambda > 0$ ensures the covariance matrix $V_t$ is positive definite.

**Optimistic reward estimate.** In the round $t$, the optimistic reward estimate/ upper confidence bound (UCB) of any context-arm feature vector $x$ is computed as follows: $\text{UCB}_{t,a}(x) \doteq \hat{\theta}_t^\top x + \alpha_t \|x\|_{V_t^{-1}}$, where $\hat{\theta}_t^\top x$ denotes the estimated reward for the context $x$ and $\alpha_t \|x\|_{V_t^{-1}}$ is the confidence bonus in which $\alpha_t \doteq R \left( d \log \left( \frac{1 + tL^2/\lambda}{\delta} \right) \right)^{\frac{1}{2}} + \lambda^{\frac{1}{2}} S$ is a slowly increasing function in $t$ and the value of $\|x\|_{V_t^{-1}}$ (i.e., weighted $l_2$-norm of vector $x$ with respect to matrix $V_t^{-1}$) goes to zero as $t$ increases.

**UCB-based algorithm.** The upper confidence bound (Li et al., 2010; Chu et al., 2011; Zhou et al., 2020) is a widely used technique for addressing the exploration-exploitation trade-off in contextual bandit problems. Our UCB-based algorithm, COBRA (UCB), for linear contextual bandit problems works as follows. At the start of round $t$ (see Fig. 2), the learner observes the context and reported arm features $x_{t,a}$, and then selects an arm $a_t = \arg\max_{a \in \mathcal{A}_t} \text{UCB}_{t,a}(x)$ (Line 5). Importantly, COBRA (UCB) does not have access to the true arm features or the true reward function parameter $\theta_\star$. As a result, over-reporting by agents can lead COBRA (UCB) to make suboptimal arm selections. To address this, we incorporate LOOM (Line 6, more details on how we adapt LOOM to linear contextual bandits are provided on the next page) to identify the over-reporting agent. By eliminating agents who satisfy the LOOM condition defined in Eq. (2) from future rounds ensures agents report truthfully.

---

**COBRA** Algorithm for **CO**ntextual **B**andits with St**RA**tegic Agents

---

1: **Input:** $\mathcal{N}_1$: set of agents before the round $t = 1$, $\delta \in (0, 1)$, and $\lambda > 0$
2: **for** $t = 1, 2, \ldots$ **do**
3:   Observe context $x_t$ and then a receive set of arm's features $\mathcal{A}_t$ reported by agents in $\mathcal{N}_t$.
4:   Select an arm $a_t = \text{argmax}_{a \in \mathcal{A}_t} \text{UCB}_{t,a}(x) \doteq \hat{\theta}_t^\top x + \alpha_t \|x\|_{V_t^{-1}}$
5:   Observe noisy reward $y_t$.
6:   Check LOOM condition in Eq. (2) for each agent in $\mathcal{N}_t$. If it holds for any agent $a$, then update $\mathcal{N}_{t+1} = \mathcal{N}_t \setminus \{a\}$.
7:   If $N_{t+1} = \emptyset$, stop and receive 0 reward thereafter.
8: **end for**

---

**TS-based algorithm.** Motivated by the empirical advantages of Thompson Sampling (TS) over UCB-based bandit algorithms (Chapelle and Li, 2011; Agrawal and Goyal, 2013; Zhang et al., 2021), we also propose a TS-based variant, COBRA (TS). This algorithm closely mirrors COBRA (UCB), differing only in the arm selection step (Line 5). To obtain a TS-based reward estimate, the algorithm first samples a reward function parameter $\tilde{\theta}_t \sim \mathbb{N}\left(\hat{\theta}_t, \beta_t^2 V_t^{-1}\right)$, where $\mathbb{N}$ denotes the normal distribution and $\beta_t = R\sqrt{9d \log(t/\delta)}$ (Agrawal and Goyal, 2013). Using $\tilde{\theta}_t$, the TS-based reward estimate, i.e., $\text{TS}_t(x_{t,a}) = x_{t,a}^\top \tilde{\theta}_t$, replaces $\text{UCB}_t(x_{t,a})$ when computing the optimistic reward in Line 5. We compare and demonstrate superior empirical performance of COBRA (TS) in Section 5.

**LOOM in COBRA (UCB) and COBRA (TS).** To check the LOOM condition defined in Eq. (2), we need to compute $\text{LCB}_{t,a}$, which requires estimating the reward function parameters using observations from all agents except agent $a$. To construct the aforementioned estimate, we exclude the observations from agent $a$, which is given as follows: $\hat{\theta}_{t,-a} = V_{t,-a}^{-1} \sum_{s=1, a_s \neq a}^{t} x_{s,a_s} y_s$, where $V_{t,-a} = \lambda I_d + \sum_{s=1, a_s \neq a}^{t} x_{s,a_s} x_{s,a_s}^\top$. We now formally define the pessimistic estimate of the total expected reward for an agent $a$ as: $\text{LCB}_{t,a}^{(x)} = \sum_{s=1, a_s=a}^{t} \text{LCB}_{t,-a}(x_{s,a_s})$, where $\text{LCB}_{t,-a}(x_{s,a_s}) = x_{s,a_s}^\top \hat{\theta}_{t,-a} - \alpha_{t,-a} \|x_{s,a_s}\|_{V_{t,-a}^{-1}}$ for $1 \leq s \leq t$, with $\alpha_{t,-a} = R\left(d \log\left(\frac{1+(t+1-S_t(a))L^2/\lambda}{\delta}\right)\right)^{\frac{1}{2}} + \lambda^{\frac{1}{2}} S$ and $S_t(a)$ denoting the number of times agent $a$ has been selected up to round $t$. Using the upper bound $\text{UCB}_{t,a}^{(y)} = \sum_{s \leq t, a_s=a} y_s + \sqrt{2 S_t(a) \log(1/\delta_{t,a}^y)}$ from Lemma 1, we can apply LOOM condition to identify whether any agent is over-reporting their arm features.

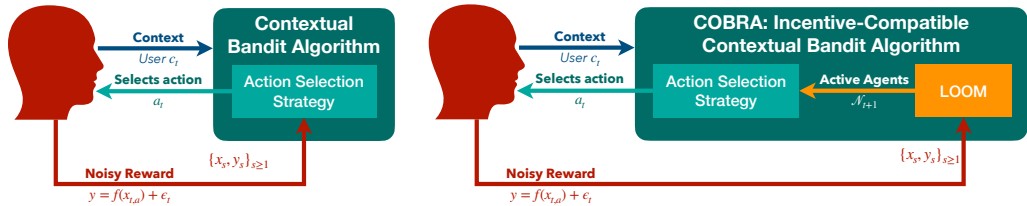

(a) Standard Contextual Bandit Algorithm     (b) COBRA: Incentive Compatible Contextual Bandit Algorithm

Figure 2: COBRA integrates LOOM as a post-processing step after each interaction round to identify over-reporting agents in a LOOM-compatible contextual bandit algorithm.

**Non-linear reward function.** We now consider contextual bandit problems with potentially non-linear reward functions. COBRA naturally generalizes to this setting by adopting any suitable *LOOM-compatible* contextual bandit algorithm to get optimistic reward estimates for context–arm feature vectors. These estimates are then used to select the best arm for a given context (as in Line 5), after which LOOM is applied as a post-processing step to identify over-reporting agents.

### 4.1 NE Guarantee and Regret Analysis

In this section, we derive NE and regret guarantees for COBRA and establish its desirable properties, including incentive compatibility (i.e., reporting truthfully is the dominant strategy) and a sublinear regret guarantee. We assume that the agent only over-reports their arm features so that the corresponding reward is higher, i.e., for all $x^\star \in \mathcal{X}$: $f(x) > f(x^\star)$, where $x$ is the reported arm features for the

true arm feature $x^\star$. Notably, we impose no restrictions on how agents report their arm features, aside from no collusion assumption, which is common in VCG-type mechanisms (Vickrey, 1961; Clarke, 1971; Groves, 1973). Let $\tilde{O}$ hide the logarithmic factors and constants. Our next result shows that when arms report truthfully, COBRA approximately incentivizes truthful behavior and achieves a regret bound of at most $\tilde{O}(d\sqrt{T})$ under this approximate NE. Next, we present the results for the linear reward function and then for the non-linear reward function.

**Theorem 2** (Linear). *When agents report truthfully, being truthful is a $\tilde{O}(d\sqrt{T})$-NE under COBRA. The regret of COBRA under this approximate NE is at most $\mathfrak{R}_T(COBRA, \boldsymbol{\sigma}^\star) = \tilde{O}(d\sqrt{T})$.*

Let $\tilde{d}$ be the effective dimension associated with contextual bandit problems with non-linear reward functions. Let $\mathfrak{A}$ be a LOOM-compatible contextual bandit algorithm for which $|f_t(x) - f(x)| \leq h(x, \mathcal{O}_t)$ holds with probability at least $1 - \delta$ for any $x \in \mathcal{X}$ and $\sqrt{\sum_{t=1}^{T} [h(x_{t,a_t}, \mathcal{O}_t)]^2} = \tilde{O}(\tilde{d}\log T)$. For notational simplicity, we assume that this bound holds for the algorithm $\mathfrak{A}$ used by COBRA.

**Theorem 3** (Non-linear). *Let $\mathfrak{A}$ be a LOOM-compatible contextual bandit algorithm used by COBRA. When agents report truthfully, being truthful is a $\tilde{O}(\tilde{d}\sqrt{T})$-NE under COBRA. With probability at least $1 - \delta_x - \delta_y$, the regret of COBRA under this approximate NE is $\mathfrak{R}_T(COBRA(\mathfrak{A}), \boldsymbol{\sigma}^\star) = \tilde{O}(\tilde{d}\sqrt{T})$.*

When multiple agents over-report, all COBRA estimators become biased (Lemma 8 in Appendix) as the over-reported arm features used for reward function estimation no longer reflect the true distribution. Our subsequent results hold only under the conditions specified in the following assumptions.

**Assumption 1.** *Let $x$ and $x^\star$ be the reported and true context-arm feature vector, respectively. Then, we assume (i) $\forall t \geq 1, a \in \mathcal{A}_t : f(x_{t,a}) \leq UCB_t(x_{t,a})$, where $UCB_t(x) = f_t(x) + h(x, \mathcal{O}_t)$. (ii) $\forall t \geq 1, a \in \mathcal{A}_t : UCB_t(x_{t,a}) \leq UCB_{t,-a}(x_{t,a})$, where $UCB_{t,-a}(x) = f_{t,-a}(x) + h(x, \mathcal{O}_{t,-a})$.*

The first part of assumption states that each agent's expected true reward for the reported features is upper bounded by the optimistic reward estimate, $UCB_t(x_{t,a})$, that uses all available context-arm features to estimate $\theta_\star$. The second assumption says that the optimistic reward estimate, when using all available context-arm features, is tighter than the optimistic reward estimate when excluding reported context-arm features of any agent. Additional discussion about these assumptions are provided in Section D. Next, we prove a strategic regret bound that holds for every NE of the agents.

**Theorem 4** (Linear). *If Assumption 1 hold then, the regret of COBRA is $\mathfrak{R}_T(COBRA, \boldsymbol{\sigma}) = \tilde{O}(d\sqrt{T} + \sqrt{NT})$ for every $\boldsymbol{\sigma} \in NE(COBRA)$. Hence, $\max_{\sigma \in NE(COBRA)} \mathfrak{R}_T(COBRA, \boldsymbol{\sigma}) = \tilde{O}(d\sqrt{T} + \sqrt{NT})$.*

Our next result extends the previous result to the general setting with non-linear reward functions.

**Theorem 5** (Non-linear). *Let $\mathfrak{A}$ be a LOOM-compatible contextual bandit algorithm used by COBRA. If Assumption 1 hold then, the regret of COBRA is $\mathfrak{R}_T(COBRA(\mathfrak{A}), \boldsymbol{\sigma}) = \tilde{O}(\tilde{d}\sqrt{T} + \sqrt{NT})$ for every $\boldsymbol{\sigma} \in NE(COBRA(\mathfrak{A}))$. Hence, $\max_{\sigma \in NE(COBRA(\mathfrak{A}))} \mathfrak{R}_T(COBRA(\mathfrak{A}), \boldsymbol{\sigma}) = \tilde{O}(\tilde{d}\sqrt{T} + \sqrt{NT})$.*

**Outline of the proofs.** The proofs of Theorem 2 and Theorem 4 depend on the LOOM mechanism to identify agents who are over-reporting. LOOM ensures that optimistic estimates are tightly bounded, thereby limiting the potential benefit from over-reporting and reinforcing truthfulness as the optimal strategy for agents. The $\sqrt{NT}$ term in Theorem 4 arises due to the strategic nature of the agents who can exploit initial noisy estimates of COBRA. The detailed proofs are provided in Section B. The proofs of Theorem 3 and Theorem 5 rely on the LOOM mechanism to identify over-reporting agents. The remainder of the proof proceeds as before, making use of Definition 4 and Assumption 1.

## 5 EXPERIMENTS

In this section, we aim to corroborate our theoretical results and empirically demonstrate the performance of our proposed algorithm in different strategic contextual bandit problems. We repeat all our experiments 20 times and show the regret (as defined in Eq. (1)) with a 95% confidence interval (the vertical line on each curve shows the confidence interval). To demonstrate the different performance aspects of our proposed algorithm, we have used different synthetic problem instances (commonly used experiment choices in bandit literature) whose details are as follows.

**Experiment setting.** We use a $d_c$-dimensional space to generate the sample features of each context, where context $c_t$ is represented by $c_t = (x_{c_t,1}, \ldots, x_{c_t,d_c})$ for $t \geq 1$. Similarly, we use a

$d_n$-dimensional space to generate the sample of each agent's arm features, where agent $n \in \mathcal{N}$ is represented by $a_t^{(n)} = \left( x_{a_t,1}^{(n)}, \ldots, x_{a_t,d_n}^{(n)} \right)$. The value of $i$-the feature $x_{c_t,i}$ (or $x_{a_t,i}^{(n)}$) is sampled uniformly at random from $(0, 2)$. To get the context-arm feature vectors for context $c_t$ in the round $t$, we concatenate the context features $c_t$ with all arm feature vectors. For context $c_t$ and agent $n$, the concatenated feature vector is denoted by $x_{t,n}$, which is an $d$-dimensional vector with $d = d_c + d_n$. We select a $d$-dimensional vector $\theta_\star$ by sampling uniformly at random from $(0, 2)^d$ and normalizing it to have unit $l_2$-norm. In all experiments, we use $\lambda = 0.01$, $R = 0.1$, $\delta = 0.05$, and $d_c = d_n$.

**Strategic over-reporting.** We consider two types of strategic manipulations: *(I) Feature adaptation:* The strategic agent updates the arm features it reports based on past selection outcomes using a finite-difference stochastic gradient ascent update. Specifically, the agent receives a binary feedback signal: $1$ if it was selected in the previous round and $0$ otherwise. Agent uses this feedback to iteratively adjust its reported features to increase the probability of being selected in future rounds. *(II) Systematic over-reporting:* To maximize the likelihood of selection, the strategic agent over-reports its feature vector according to $x = (1 + \Delta_x)x^\star$, where $x^\star$ denotes the true arm features. The agent maintains an estimate of the optimal over-reporting factor, $\hat{\Delta}_x$, which guides the extent of over-reporting. More details about these strategic manipulations are provided in Section E.

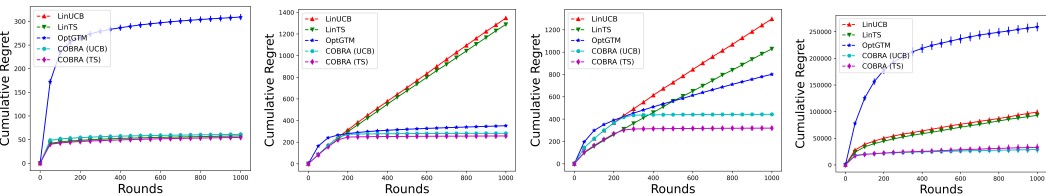

(a) Linear (Truthful setting)  (b) Linear (Agent type I)  (c) Linear (Agent type II)  (d) Square (Agent type II)

Figure 3: Comparing cumulative regret of COBRA with baselines using different problem instances.

**Regret comparison with baselines.** We compare the regret of the proposed algorithms with three baselines: Lin-UCB (Li et al., 2010), Lin-TS (Agrawal and Goyal, 2013), and OptGTM (Kleine Buening et al., 2024). For our experiments, we consider two reward functions: **Linear**, $f(x) = 5x^\top\theta_\star$, and **Square**, $f(x) = 10(x^\top\theta_\star)^2$. We use 1000 contexts, 5 agents, and $d_c = d_n = 5$, resulting in a context-arm feature dimension of $d = 10$. We evaluate four problem instances with the same setup, except using two different types of strategic over-reporting: **Agent type I:** Feature adaptation with a learning rate of $\eta = 0.05$. **Agent type II:** Systematic over-reporting, where $\hat{\Delta}_x \sim \mathbb{N}(\Delta_x^\star, \sigma_\Delta^2)$. Here, $\mathbb{N}$ denotes a normal distribution, $\Delta_x^\star$ is the optimal scaling factor such that $f((1 + \Delta_x^\star)x^\star)$ gives the highest reward among all arms, and $\sigma_\Delta$ represents the standard deviation. We assume that only one agent over-reports, and the maximum perturbation in each round is bounded by $\Delta_{\max} = 1.0$.

In Fig. 3a, all agents report truthfully under the **Linear** reward function. Even in this setting, our algorithm outperforms the state-of-the-art OptGTM and matches the performance of standard contextual bandit algorithms (LinUCB and LinTS). As expected, our proposed algorithm COBRA, based on UCB and TS variants of contextual bandits, outperforms all baselines across different problem instances with **Linear** and **Square** reward functions (Fig. 3b-3d). For the **Square** reward function, we estimate it using kernel regression with a polynomial kernel of degree 2. We observe that the TS-based variants of COBRA consistently outperform their UCB-based counterparts. Additional experimental results and ablations are provided in Section E.

## 6 CONCLUSION

This paper addresses a contextual bandit problem involving strategic agents who may misreport arm features to increase their own utility. To tackle this challenge, we propose LOOM, a mechanism that identifies over-reporting agents by leveraging the reported arm features from other agents. Building on LOOM, we introduce an algorithm, COBRA, for contextual bandit problems with strategic agents. COBRA disincentivizes strategic behavior without relying on monetary incentives, while ensuring incentive compatibility and achieving a sub-linear regret guarantee. Our experimental results across different problem instances further demonstrate the performance advantages of the proposed algorithm. A few promising directions for future work include incorporating fairness constraints into the arm selection process, developing better mechanisms capable of reliably detecting both under-reporting and over-reporting agents, and handling more complex forms of strategic behavior.

ETHICS STATEMENT

This work is primarily theoretical, focusing on the design and analysis of algorithms. The proposed methods do not directly involve human subjects, personal data, or real-world deployments. While the framework could potentially be applied in systems that interact with users, we emphasize that ethical considerations, such as fairness, privacy, and informed consent, must be addressed in practical deployments. Our primary goal is to advance the theoretical understanding of incentive-compatible contextual bandit algorithms, and we do not anticipate any immediate negative societal impacts.

REPRODUCIBILITY STATEMENT

This paper primarily presents theoretical results, including formal proofs of incentive compatibility and regret bounds. All assumptions, definitions, and derivations are stated explicitly in the main text (see Section 4) and the Appendix. The details of our experimental setup are provided in Section 5 and the Appendix. Additionally, the code used in our experiments has been included in the supplementary material, enabling full reproduction of the results reported in this paper.

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

# A APPENDIX

## A.1 ADDITIONAL RELATED WORK

**Contextual bandits.** Contextual bandits (Slivkins, 2019; Lattimore and Szepesvári, 2020) have many real-life applications, such as online recommendations, advertising, web search, and e-commerce. In this framework, a learner selects an arm and receives a reward for that choice. Given the potentially large or infinite set of arms, the mean reward for each arm is typically modeled as an unknown function, which may be linear (Li et al., 2010; Chu et al., 2011; Abbasi-Yadkori et al., 2011; Agrawal and Goyal, 2013), generalized linear model (GLM) (Filippi et al., 2010; Li et al., 2017; Jun et al., 2017; Verma et al., 2025), or non-linear (Valko et al., 2013; Chowdhury and Gopalan, 2017; Zhou et al., 2020; Zhang et al., 2021). The learner's objective is to identify the optimal action as efficiently as possible, which depends on how tightly the confidence bounds for the reward-function mapping actions to rewards are defined. Several works have explored various sources of information and side observations to enhance the learning process (Li et al., 2010; Agrawal and Goyal, 2013; Alon et al., 2015; Wu et al., 2015; Li et al., 2017; Verma and Hanawal, 2021; Verma et al., 2023).

**Strategic learning.** There are several works on strategic learning (Liu and Chen, 2016; Freeman et al., 2020; Gast et al., 2020; Zhang and Conitzer, 2021; Harris et al., 2022; 2023) and strategic classification (Hardt et al., 2016; Dong et al., 2018; Sundaram et al., 2023). The strategic classification problem was first introduced in Hardt et al. (2016). The authors considered a sequential game between a decision-maker selecting a classifier, and a strategic agent who responds by modifying their features. Chang et al. (2024) address the problem of identifying agents who exhibit the highest degree of strategic manipulation in their inputs, given a dataset of agents and their observed model inputs in a offline setting. Our work aligns with this research direction, as it explores the interaction between a strategic agent and a learning algorithm. However, unlike prior studies where agents interact with the learner only once to achieve a desired outcome, our setting involves repeated interactions, forming a repeated game without monetary transactions. Our main contribution is the development of an incentive-compatible mechanism designed to handle repeated interactions with strategic agents, specifically tailored for contextual bandit problems.

# B LEFTOVER PROOFS

## B.1 LEFTOVER PROOFS FROM SECTION 3

**Lemma 1.** *Let $S_t(a)$ be the number of times that agent $a$ is selected until round $t$, and $\varepsilon_s$ be $R$-sub-Gaussian in the observed reward $y_s$, where $1 \leq s \leq t$. Then, with probability at least $1 - \delta_{t,a}^y$*

$$\sum_{s \leq t, a_s = a} f(x^{\star}_{s,a_s}) \leq \sum_{s \leq t, a_s = a} y_s + \sqrt{2R^2 S_t(a) \log(1/\delta_{t,a}^y)}.$$

*Proof.* Recall, the observed reward in round $t$ is $y_t = f(x^{\star}_{t,a_t}) + \varepsilon_t$, where $\varepsilon_t$ is $R$-sub-Gaussian noise. We want to get the upper bound for the sum of observed rewards in terms of the sum of true rewards, i.e., $\sum_{s \leq t, a_s = a} \left( y_s - f(x^{\star}_{s,a_s}) \right)$. Note that $\varepsilon_s = y_s - f(x^{\star}_{s,a_s})$ is a $R$-sub-Gaussian random variable. Using Hoeffding inequality for the sum of sub-Gaussian random variables, we get

$$\text{For any } \tau > 0, \ \mathbb{P}\left\{ \sum_{s \leq t, a_s = a} \varepsilon_s \geq \tau \right\} \leq \exp\left( -\frac{\tau^2}{2R^2 S_t(a)} \right).$$

Setting $\tau = \sqrt{2R^2 S_t(a) \log(1/\delta_{t,a}^y)}$, we get

$$\mathbb{P}\left\{ \sum_{s \leq t, a_s = a} \varepsilon_s \geq \sqrt{2R^2 S_t(a) \log(1/\delta_{t,a}^y)} \right\} \leq \delta_{t,a}^y.$$

Expanding $\varepsilon_s = y_s - f(x^{\star}_{s,a_s})$ in the above equation, we can have the following results with probability at least $1 - \delta_{t,a}^y$,

$$\sum_{s \leq t, a_s = a} y_s \geq \sum_{s \leq t, a_s = a} f(x^{\star}_{s,a_s}) - \sqrt{2R^2 S_t(a) \log(1/\delta_{t,a}^y)}. \tag{3}$$

Similarly, with with probability at least $1 - \delta_{t,a}^y$,

$$\sum_{s \leq t, a_s = a} y_s \leq \sum_{s \leq t, a_s = a} f(x_{s,a_s}^\star) + \sqrt{2R^2 S_t(a) \log(1/\delta_{t,a}^y)}. \tag{4}$$

After re-arrangements of some terms in Eq. (3), the sum of true rewards must be less than the upper bound of observed rewards with probability at least $1 - \delta_{t,a}^y$, i.e.,

$$\sum_{s \leq t, a_s = a} f(x_{s,a_s}^\star) \leq \sum_{s \leq t, a_s = a} y_s + \sqrt{2R^2 S_t(a) \log(1/\delta_{t,a}^y)}. \qquad \square$$

**Theorem 1.** *Let agent $a$ reports truthfully. Then, LOOM does not eliminate agent $a$ with high probability at least $1 - \delta_{t,a}^x - \delta_{t,a}^y$.*

*Proof.* Since all agents report truthfully, for all $t \geq 1, a \in \mathcal{A}_t : x^\star = x$. Note that we are estimating the reward function $f$ using available observations observed context-arm features and rewards. Recall that we use $\mathcal{O}_{t,-a}$ to denote the observations from all agents except agent $a$ and $f_{t,-a}$ represents the estimate of reward function $f$ using $\mathcal{O}_{t,-a}$ at the end of round $t$. Even if other agents report truthfully, noisy reward feedback may lead to an inaccurate estimator. Let the confidence ellipsoid $|f_{t,-a}(x) - f(x)| \leq h(x, \mathcal{O}_{t,-a})$ hold with probability $1 - \delta_{t,a}$. Then, for any $x \in \mathcal{X}$, $\text{LCB}_{t,-a}(x) = f_{t,-a}(x) - h(x, \mathcal{O}_{t,-a})$ is the pessimistic estimates of the expected reward for $x$ that also holds with probability $1 - \delta_{t,a}$. Furthermore, $f(x) \geq \text{LCB}_{t,-a}(x)$ (see Lemma 6 in Section C.1 for more details). Using this, for any $x_{t,a_t} \in \mathcal{X}$, we have

$$f(x_{t,a_t}^\star) = f(x_{t,a_t}) \geq \text{LCB}_{t,-a}(x_{t,a_t}) \implies f(x_{t,a_t}^\star) \geq \text{LCB}_{t,-a}(x_{t,a_t}).$$

Next, we can lower bound the sum of true rewards in terms of the lower confidence bound on estimated rewards using observed context-arm feature vectors as follows:

$$\sum_{s \leq t, a_s = a} f(x_{s,a_a}^\star) \geq \sum_{s \leq t, a_s = a} \text{LCB}_{t,-a}(x_{s,a_s})$$

$$\implies \sum_{s \leq t, a_s = a} \text{LCB}_{t,-a}(x_{s,a_s}) \leq \sum_{s \leq t, a_s = a} f(x_{s,a_a}^\star). \tag{5}$$

For brevity, we assume the above bound holds with probability at least $1 - \delta_{t,a}^x$ in the round $t$. Note that $\delta_{t,a}^x$ can be computed exactly when applying the union bound. Since the true reward is unknown, we instead first use the upper bound provided in Lemma 1, which holds with probability at least $\delta_{t,a}^y$, to modify Eq. (5). We then use the definitions of $\text{LCB}_{t,a}^{(x)}$ and $\text{UCB}_{t,a}^{(y)}$ to get:

$$\sum_{s \leq t, a_s = a} \text{LCB}_{t,-a}(x_{s,a_s}) \leq \sum_{s \leq t, a_s = a} y_s + \sqrt{2R^2 S_t(a) \log(1/\delta_{t,a}^y)}$$

$$\implies \text{LCB}_{t,a}^{(x)} \leq \text{UCB}_{t,a}^{(y)}. \tag{6}$$

If the sum of the lower bound of estimated rewards is less than the upper bound of observed rewards for an agent then that agent is not mis-reporting. However, if any agent violates Eq. (6), i.e., $\text{LCB}_{t,a}^{(x)} > \text{UCB}_{t,a}^{(y)}$, then that agent is not truthful. The probability of failing this LOOM condition is upper bounded by $\delta_{t,a}^x + \delta_{t,a}^y$. Since this condition is used as a criterion in COBRA to identify the strategic agent, COBRA does not eliminate a truthful agent with probability at least $1 - \delta_{t,a}^x - \delta_{t,a}^y$. $\square$

### B.2 Leftover proofs from Section 4

The following lemmas are fundamental to the proof of our theoretical results. We follow the following notation throughout the proof: the arm is represented by $a$, and $-a$ represents other than arm $a$'s estimate. We use $\|x\|_A$ to denote the weighted $l_2$-norm of vector $x$ with respect to matrix $A$. We next state the following result that gives the confidence ellipsoid with center at $\hat{\theta}_t$ or confidence set for the case when the reward function is linear. We will use this result to prove our bounds in Section 4.

**Lemma 2.** *Let $\delta \in (0,1)$, $\lambda > 0$, $R > 0$, $\hat{\theta}_t = V_t^{-1} \sum_{s=1}^{t-1} x_{s,a_s} y_s$, $V_t = \lambda I + \sum_{s=1}^{t-1} x_{s,a_s} x_{s,a_s}^\top$. Then, with probability at least $1 - \delta$, for all $t \geq 1$, $\theta_\star$ lies in the following confidence set:*

$$
C_t = \left\{ \theta \in \mathbb{R}^d \colon \left\| \hat{\theta}_t - \theta \right\|_{V_t} \leq \alpha_t \right\}, \text{ where } \alpha_t = \left( R \sqrt{d \log \left( \frac{1 + (tL^2/\lambda)}{\delta} \right)} + \lambda^{\frac{1}{2}} S \right).
$$

*Furthermore, with probability at least $1 - \delta$,*

$$
\forall x \in \mathcal{X} : \theta_\star^\top x \leq UCB_t(x) = \hat{\theta}_t^\top x + \alpha_t \left\| x \right\|_{V_t^{-1}}.
$$

*Similarly, with probability at least $1 - \delta$,*

$$
\forall x \in \mathcal{X} : \theta_\star^\top x \geq LCB_t(x) = \hat{\theta}_t^\top x - \alpha_t \left\| x \right\|_{V_t^{-1}}.
$$

*Proof.* The proof of the first part of the results directly follows from Theorem 2 of Abbasi-Yadkori et al. (2011). The proof of the second part follows from the first part with some simple algebraic simplifications as follows:

$$
\theta_\star^\top x - \hat{\theta}_t^\top x \leq |\hat{\theta}_t^\top x - \theta_\star^\top x|
$$
$$
\implies \theta_\star^\top x - \hat{\theta}_t^\top x \leq \left\| \hat{\theta}_t - \theta_\star \right\|_{V_t} \left\| x \right\|_{V_t^{-1}}
$$
$$
\implies \theta_\star^\top x \leq \hat{\theta}_t^\top x + \alpha_t \left\| x \right\|_{V_t^{-1}}
$$
$$
\implies \theta_\star^\top x \leq \text{UCB}_t(x).
$$

Similarly, the last part also follows from the first part with some simple algebraic simplifications as follows:

$$
|\hat{\theta}_t^\top x - \theta_\star^\top x| \leq \left\| x \right\|_{V_t^{-1}} \left\| \hat{\theta}_t - \theta_\star \right\|_{V_t}.
$$

After reversing the above inequality, we have

$$
\left\| x \right\|_{V_t^{-1}} \left\| \hat{\theta}_t - \theta_\star \right\|_{V_t} \geq |\hat{\theta}_t^\top x - \theta_\star^\top x| \geq \hat{\theta}_t^\top x - \theta_\star^\top x
$$
$$
\implies \left\| x \right\|_{V_t^{-1}} \left\| \hat{\theta}_t - \theta_\star \right\|_{V_t} \geq \hat{\theta}_t^\top x - \theta_\star^\top x
$$
$$
\implies \theta_\star^\top x \geq \hat{\theta}_t^\top x - \left\| \hat{\theta}_t - \theta_\star \right\|_{V_t} \left\| x \right\|_{V_t^{-1}}
$$
$$
\implies \theta_\star^\top x \geq \hat{\theta}_t^\top x - \alpha_t \left\| x \right\|_{V_t^{-1}}
$$
$$
\implies \theta_\star^\top x \geq \text{LCB}_t(x). \qquad \square
$$

Note that it is possible $\theta_\star$ may not belong to the confidence ellipsoid of $\theta$. However, when all agents are truthful, i.e., $x = x^\star$, thereby $\theta = \theta_\star$ is trivially satisfied. Recall the following definitions from the main paper (note that we estimated the ordinary least square (OLS) closed-form solution excluding the information of agent $a$):

$$
\hat{\theta}_{t,-a} = V_{t,-a}^{-1} \sum_{s=1, a_s \neq a}^{t-1} x_{s,a_s} y_s, \text{ with } V_{t,-a} = \lambda I + \sum_{s=1, a_s \neq a}^{t-1} x_{s,a_s} x_{s,a_s}^\top.
$$

**Lemma 3.** *Let $\delta \in (0,1)$, $\lambda > 0$, and $R > 0$. Then, with probability $1 - \delta$,*

$$
\left\| \hat{\theta}_{t,-a} - \theta_\star \right\|_{V_{t,-a}} \leq \left( R \sqrt{d \log \left( \frac{1 + (t - S_t(a))L^2/\lambda}{\delta} \right)} + \lambda^{\frac{1}{2}} S \right) = \alpha_{t,-a}.
$$

*Furthermore, with probability at least $1 - \delta$, the upper bound of $\theta_\star^\top x$ is given by*

$$
\forall x \in \mathcal{X} : \theta_\star^\top x \leq UCB_{t,-a}(x) = \hat{\theta}_{t,-a}^\top x + \alpha_{t,-a} \left\| x \right\|_{V_{t,-a}^{-1}}.
$$

*Similarly, with probability at least $1 - \delta$, the lower bound of $\theta_\star^\top x$ is given by*

$$
\forall x \in \mathcal{X} : \theta_\star^\top x \geq LCB_{t,-a}(x) = \hat{\theta}_{t,-a}^\top x - \alpha_{t,-a} \left\| x \right\|_{V_{t,-a}^{-1}}.
$$

*Proof.* The first part of the proof follows from Lemma 2 as we are not using observations associated with agent $a$, reducing to the standard confidence bound restricted to observations of all agents except $a$. The proof of the second part follows from the first part with some simple algebraic simplifications as follows:

$$\theta_\star^\top x - \hat{\theta}_{t,-a}^\top x \le |\hat{\theta}_{t,-a}^\top x - \theta_\star^\top x|$$

$$\implies \theta_\star^\top x - \hat{\theta}_{t,-a}^\top x \le \left\| \hat{\theta}_{t,-a} - \theta_\star \right\|_{V_{t,-a}} \|x\|_{V_{t,-a}^{-1}}$$

$$\implies \theta_\star^\top x \le \hat{\theta}_{t,-a}^\top x + \alpha_{t,-a} \|x\|_{V_{t,-a}^{-1}}$$

$$\implies \theta_\star^\top x \le \mathrm{UCB}_{t,-a}(x).$$

Similarly, the last part follows from the first part with some algebraic simplifications as follows:

$$|\hat{\theta}_{t,-a}^\top x - \theta_\star^\top x| \le \|x\|_{V_{t,-a}^{-1}} \left\| \hat{\theta}_{t,-a} - \theta_\star \right\|_{V_{t,-a}}.$$

After reversing the above inequality, we have

$$\|x\|_{V_{t,-a}^{-1}} \left\| \hat{\theta}_{t,-a} - \theta_\star \right\|_{V_{t,-a}} \ge |\hat{\theta}_{t,-a}^\top x - \theta_\star^\top x| \ge \hat{\theta}_{t,-a}^\top x - \theta_\star^\top x$$

$$\implies \|x\|_{V_{t,-a}^{-1}} \left\| \hat{\theta}_{t,-a} - \theta_\star \right\|_{V_{t,-a}} \ge \hat{\theta}_{t,-a}^\top x - \theta_\star^\top x$$

$$\implies \theta_\star^\top x \ge \hat{\theta}_{t,-a}^\top x - \left\| \hat{\theta}_{t,-a} - \theta_\star \right\|_{V_{t,-a}} \|x\|_{V_{t,-a}^{-1}}$$

$$\implies \theta_\star^\top x \ge \hat{\theta}_{t,-a}^\top x - \alpha_{t,-a} \|x\|_{V_{t,-a}^{-1}}$$

$$\implies \theta_\star^\top x \ge \mathrm{LCB}_{t,-a}(x). \qquad \square$$

### B.2.1 PROOF OF THEOREM 2

**Theorem 2** (Linear). *When agents report truthfully, being truthful is a $\tilde{O}(d\sqrt{T})$-NE under COBRA. The regret of COBRA under this approximate NE is at most $\mathfrak{R}_T(COBRA, \boldsymbol{\sigma}^\star) = \tilde{O}(d\sqrt{T})$.*

*Proof.* When all agents report truthfully, our algorithm is the same as Lin-UCB (Chu et al., 2011) with a mechanism for identifying strategic agents that holds with probability $1 - \delta_x - \delta_y$. For completeness, we first prove the regret upper bound of COBRA as follows:

$$\mathfrak{R}_T(\mathrm{COBRA}, \boldsymbol{\sigma}^\star) = \sum_{t=1}^T (\theta_\star^\top x_{t,a_t^\star}^\star - \theta_\star^\top x_{t,a_t}^\star). \tag{7}$$

Since the true feature vector is the same as the reported context-arm feature vector (i.e., $x_{t,a}^\star = x_{t,a}$), we can start with upper bounding the difference $\theta_\star^\top x_{t,a_t^\star}^\star - \theta_\star^\top x_{t,a_t}^\star$ as follows:

$$\theta_\star^\top x_{t,a_t^\star}^\star - \theta_\star^\top x_{t,a_t}^\star = \theta_\star^\top x_{t,a_t^\star} - \theta_\star^\top x_{t,a_t}$$

$$\le \mathrm{UCB}(x_{t,a_t^\star}) - \theta_\star^\top x_{t,a_t}$$

$$\le \mathrm{UCB}(x_{t,a_t}) - \theta_\star^\top x_{t,a_t} \quad \left(\text{as } \mathrm{UCB}(x_{t,a_t^\star}) \le \mathrm{UCB}(x_{t,a_t})\right)$$

$$= \hat{\theta}_t^\top x_{t,a_t} + \alpha_t \|x_{t,a_t}\|_{V_t^{-1}} - \theta_\star^\top x_{t,a_t}$$

$$= \hat{\theta}_t^\top x_{t,a_t} - \theta_\star^\top x_{t,a_t} + \alpha_t \|x_{t,a_t}\|_{V_t^{-1}}$$

$$\le \left\| \theta_\star - \hat{\theta}_t \right\|_{V_t} \|x_{t,a_t}\|_{V_t^{-1}} + \alpha_t \|x_{t,a_t}\|_{V_t^{-1}}$$

$$\le \alpha_t \|x_{t,a_t}\|_{V_t^{-1}} + \alpha_t \|x_{t,a_t}\|_{V_t^{-1}}$$

$$\implies \theta_\star^\top x_{t,a_t^\star}^\star - \theta_\star^\top x_{t,a_t}^\star \le 2\alpha_t \|x_{t,a_t}\|_{V_t^{-1}}. \tag{8}$$

Note that $\hat{\theta}_t$ is an estimator of $\theta_\star$ as the true feature vector is the same as the reported context-arm feature vector. After using the upper bound given in Eq. (8) into Eq. (7), we get an upper bound on the regret as follows:

$$\mathfrak{R}_T\left(\text{COBRA}, \boldsymbol{\sigma}^\star\right) = \theta_\star^\top x_{t,a_t^\star}^\star - \theta_\star^\top x_{t,a_t}^\star$$

$$\leq \sum_{t=1}^{T} 2\alpha_t \left\|x_{t,a_t}\right\|_{V_t^{-1}}$$

$$= 2\sum_{t=1}^{T} \alpha_t \left\|x_{t,a_t}\right\|_{V_t^{-1}}$$

$$\leq 2\sqrt{T}\sqrt{\sum_{t=1}^{T}\left[\alpha_t \left\|x_{t,a_t}\right\|_{V_t^{-1}}\right]^2}$$

$$\leq 2\sqrt{T}\sqrt{\sum_{t=1}^{T}\left[\alpha_T \left\|x_{t,a_t}\right\|_{V_t^{-1}}\right]^2}$$

$$= 2\sqrt{T}\sqrt{\alpha_T^2 \sum_{t=1}^{T}\left\|x_{t,a_t}\right\|_{V_t^{-1}}^2}$$

$$= 2\alpha_T\sqrt{T}\sqrt{\sum_{t=1}^{T}\left\|x_{t,a_t}\right\|_{V_t^{-1}}^2}$$

$$\leq 2\alpha_T\sqrt{T}\sqrt{2\log\frac{\det(V_T)}{\det(\lambda I_d)}}$$

$$\implies \mathfrak{R}_T\left(\text{COBRA}, \boldsymbol{\sigma}^\star\right) \leq 2\alpha_T\sqrt{2dT\log(\lambda + TL/d)} = \tilde{O}(d\sqrt{T}). \tag{9}$$

The first inequality directly follows from Eq. (8). The second inequality is due to using Cauchy-Schwarz inequality where third inequality follows from the fact that $\alpha_t$ increases with $t$. The last two inequalities follow from Lemma 11 and Lemma 10 of Abbasi-Yadkori et al. (2011), respectively, and $\alpha_T = \tilde{O}(d\log T)$.

We now prove that being truthful is an approximate Nash equilibrium for COBRA. Recall, $S_T(a)$ denotes the number of times an agent being selected by COBRA, which is given as follows:

$$S_T(a) = \sum_{t=1}^{T} \mathbb{1}(a_t = a)$$

$$= \sum_{t=1}^{T} \mathbb{1}(a_t = a, a_t^\star = a) + \sum_{t=1}^{T} \mathbb{1}(a_t = a, a_t^\star \neq a)$$

$$\geq \sum_{t=1}^{T} \mathbb{1}(a_t^\star = a) - \sum_{t=1}^{T} \mathbb{1}(a_t^\star = a, a_t \neq a)$$

$$\geq \sum_{t=1}^{T} \mathbb{1}(a_t^\star = a) - \sum_{t=1}^{T} \mathbb{1}(a_t \neq a_t^\star)$$

$$\implies S_T(a) \geq S_T^\star(a) - \sum_{t=1}^{T} \mathbb{1}(a_t \neq a_t^\star). \tag{10}$$

To get the lower bound $S_T(a)$, we get an upper bound $\sum_{t=1}^{T} \mathbb{1}(a_t \neq a_t^\star)$. Let $\Delta_{a_t} = \left(\theta_\star^\top x_{t,a_t^\star}^\star - \theta_\star^\top x_{t,a_t}^\star\right) > 0$ for $a_t \neq a_t^\star$. We multiply and divide $\mathbb{1}(a_t \neq a_t^\star)$ by $\Delta_{a_t}$ and then

use inequality in Eq. (8), i.e., $\Delta_{a_t} \leq 2\alpha_t \|x_{t,a_t}\|_{V_t^{-1}}$ as follows:

$$\sum_{t=1}^{T} \mathbb{1}(a_t \neq a_t^\star) = \sum_{t=1}^{T} \mathbb{1}(a_t \neq a_t^\star) \frac{\Delta_{a_t}}{\Delta_{a_t}}$$

$$\leq \sum_{t=1}^{T} \mathbb{1}(a_t \neq a_t^\star) \frac{2\alpha_t \|x_{t,a_t}\|_{V_t^{-1}}}{\Delta_{a_t}} \quad (\text{as } x_{t,a}^\star = x_{t,a})$$

$$\leq \sum_{t=1}^{T} \frac{2\alpha_t \|x_{t,a_t}\|_{V_t^{-1}}}{\Delta_{a_t}}$$

$$= \sum_{t=1}^{T} \frac{1}{\Delta_{a_t}} 2\alpha_t \|x_{t,a_t}\|_{V_t^{-1}}$$

$$\leq \sqrt{\sum_{t=1}^{T} \left(\frac{1}{\Delta_{a_t}}\right)^2 \sum_{t=1}^{T} \left(2\alpha_t \|x_{t,a_t}\|_{V_t^{-1}}\right)^2}$$

$$\leq \sqrt{\sum_{t=1}^{T} \left(\frac{1}{\Delta_{a_t}}\right)^2 \sum_{t=1}^{T} \left(2\alpha_T \|x_{t,a_t}\|_{V_t^{-1}}\right)^2}$$

$$= \sqrt{\sum_{t=1}^{T} \left(\frac{1}{\Delta_{a_t}}\right)^2} \sqrt{\sum_{t=1}^{T} \left(2\alpha_T \|x_{t,a_t}\|_{V_t^{-1}}\right)^2}$$

$$= \sqrt{\sum_{t=1}^{T} \left(\frac{1}{\Delta_{a_t}}\right)^2} \sqrt{(2\alpha_T)^2 \sum_{t=1}^{T} \|x_{t,a_t}\|_{V_t^{-1}}^2}$$

$$= 2\alpha_T \sqrt{\sum_{t=1}^{T} \left(\frac{1}{\Delta_{a_t}}\right)^2} \sqrt{\sum_{t=1}^{T} \|x_{t,a_t}\|_{V_t^{-1}}^2}$$

$$\leq 2\alpha_T \sqrt{\sum_{t=1}^{T} \left(\frac{1}{\Delta_{a_t}}\right)^2} \sqrt{2\log \frac{\det(V_T)}{\det(\lambda I_d)}}$$

$$\leq 2\alpha_T \sqrt{\sum_{t=1}^{T} \left(\frac{1}{\Delta_{a_t}}\right)^2} \sqrt{2d\log(\lambda + TL/d)}$$

$$\leq 2\alpha_T \sqrt{\sum_{t=1}^{T} \left(\frac{1}{\Delta_{\min}}\right)^2} \sqrt{2d\log(\lambda + TL/d)}$$

$$\leq \frac{2}{\Delta_{\min}} \alpha_T \sqrt{T} \sqrt{2d\log(\lambda + TL/d)}$$

$$\implies \sum_{t=1}^{T} \mathbb{1}(a_t \neq a_t^\star) \leq \frac{2}{\Delta_{\min}} \left(R\sqrt{d\log\left(\frac{1 + (tL^2/\lambda)}{\delta}\right)} + \lambda^{\frac{1}{2}}S\right) \sqrt{2dT\log(\lambda + TL/d)}.$$

Note that $\Delta_{\min} = \min_{a_t \neq a_t^\star} \Delta_{a_t}$. Although using $\Delta_{\min}$ loosen the upper bound, we use this to get dependence on $T$. Let $\tilde{O}$ hide the dependence on logarithmic terms, then we have the following result:

$$\sum_{t=1}^{T} \mathbb{1}(a_t \neq a_t^\star) \leq \tilde{O}\left(d\sqrt{T}\right). \tag{11}$$

Using this upper bound in Eq. (10), we get the following bound for any agent $a \in \mathcal{A}$:

$$S_T(a) \geq S_T^\star(a) - \tilde{O}\left(d\sqrt{T}\right). \tag{12}$$

Now we consider the case where an agent $a$ deviates from the truthful strategy. The number of times an agent being selected by COBRA is given as follows:

$$
\begin{aligned}
S_T(a) &= \sum_{t=1}^{T} \mathbb{1}(a_t = a) \\
&= \sum_{t=1}^{T} \mathbb{1}(a_t = a, a_t^\star = a) + \sum_{t=1}^{T} \mathbb{1}(a_t = a, a_t^\star \neq a) \\
&\leq \sum_{t=1}^{T} \mathbb{1}(a_t^\star = a) + \sum_{t=1}^{T} \mathbb{1}(a_t = a, a_t^\star \neq a) \\
&\leq \sum_{t=1}^{T} \mathbb{1}(a_t^\star = a) + \sum_{t=1}^{T} \mathbb{1}(a_t \neq a_t^\star).
\end{aligned}
\tag{13}
$$

Using Eq. (11) in Eq. (13), we get

$$
S_T(a) \leq S_T^\star(a) + \tilde{O}\left(d\sqrt{T}\right).
\tag{14}
$$

Combining Eq. (12) and Eq. (14) completes our proof that COBRA is $\tilde{O}(d\sqrt{T})$-NE. $\qquad\square$

### B.2.2 Proof of Theorem 4

To prove Theorem 4, we will use the following result that upper bounds the total amount of regret that an agent $a$ can exert before being identified by LOOM.

**Lemma 4.** *Let* $UCB_{t,-a}(x_{s,a}) = \hat{\theta}_{t,-a}^\top x_{s,a} + \alpha_{t,-a}\|x_{s,a}\|_{V_{t,-a}^{-1}}$*, where $x_{s,a}$ is the arm feature vector associated with agent $a$ in the round $s$. Then, with probability at least* $1 - \delta_{t,a}^x - \delta_{t,a}^y$*,*

$$
\sum_{s \leq t:\, a_s = a} \left(UCB_{t,-a}(x_{s,a}) - \theta_\star^\top x_{s,a}^\star\right) \leq \sum_{s \leq t, a_s = a} 2\alpha_{t,-a}\|x_{s,a}\|_{V_{t,-a}^{-1}} + 2\sqrt{2R^2 S_t(a)\log(1/\delta_{t,a}^y)}.
$$

*Proof.* Using Eq. (6) with Lemma 3 for linear reward function that holds with probability at least $1 - \delta_{t,a}^x$, we get:

$$
\begin{aligned}
\sum_{s \leq t, a_s = a} \left(\hat{\theta}_{t,-a}^\top x_{s,a} - \alpha_{t,-a}\|x_{s,a}\|_{V_{t,-a}^{-1}}\right) &\leq \sum_{s \leq t, a_s = a} y_s + \sqrt{2R^2 S_t(a)\log(1/\delta_{t,a}^y)} \\
&\leq \sum_{s \leq t, a_s = a} \theta_\star^\top x_{s,a}^\star + \sqrt{2R^2 S_t(a)\log(1/\delta_{t,a}^y)} \\
&\quad + \sqrt{2R^2 S_t(a)\log(1/\delta_{t,a}^y)} \\
\implies \sum_{s \leq t, a_s = a} \left(\hat{\theta}_{t,-a}^\top x_{s,a} - \theta_\star^\top x_{s,a}^\star\right) &\leq \sum_{s \leq t, a_s = a} \alpha_{t,-a}\|x_{s,a}\|_{V_{t,-a}^{-1}} + 2\sqrt{2R^2 S_t(a)\log(1/\delta_{t,a}^y)}.
\end{aligned}
$$

The second inequality follows from Eq. (4) by using upper bound (as the reward function is linear) on $\sum_{s \leq t, a_s = a} y_s$ that holds with probability $1 - \delta_{t,a}^y$. Now we prove the second part of the result by replacing $\hat{\theta}_{t,-a}^\top x_{s,a}$ by $UCB_{t,-a}(x_{s,a}) - \alpha_{t,-a}\|x_{s,a}\|_{V_{t,-a}^{-1}}$ and we get

$$
\begin{aligned}
\sum_{s \leq t, a_s = a} &\left(UCB_{t,-a}(x_{s,a}) - \alpha_{t,-a}\|x_{s,a}\|_{V_{t,-a}^{-1}} - \theta_\star^\top x_{s,a}^\star\right) \\
&\leq \sum_{s \leq t, a_s = a} \alpha_{t,-a}\|x_{s,a}\|_{V_{t,-a}^{-1}} + 2\sqrt{2R^2 S_t(a)\log(1/\delta_{t,a}^y)} \\
\implies \sum_{s \leq t, a_s = a} &\left(UCB_{t,-a}(x_{s,a}) - \theta_\star^\top x_{s,a}^\star\right) \\
&\leq \sum_{s \leq t, a_s = a} 2\alpha_{t,-a}\|x_{s,a}\|_{V_{t,-a}^{-1}} + 2\sqrt{2R^2 S_t(a)\log(1/\delta_{t,a}^y)}. \qquad\square
\end{aligned}
$$

We first restate the main assumptions needed to prove Theorem 4.

**Assumption 1.** *Let $x$ and $x^\star$ be the reported and true context-arm feature vector, respectively. Then, we assume (i) $\forall t \geq 1, a \in \mathcal{A}_t : f(x_{t,a}) \leq UCB_t(x_{t,a})$, where $UCB_t(x) = f_t(x) + h(x, \mathcal{O}_t)$.*
*(ii) $\forall t \geq 1, a \in \mathcal{A}_t : UCB_t(x_{t,a}) \leq UCB_{t,-a}(x_{t,a})$, where $UCB_{t,-a}(x) = f_{t,-a}(x) + h(x, \mathcal{O}_{t,-a})$.*

We now have all results that will be used to prove Theorem 4.

**Theorem 4** (Linear)**.** *If Assumption 1 hold then, the regret of COBRA is $\mathfrak{R}_T(COBRA, \boldsymbol{\sigma}) = \tilde{O}(d\sqrt{T} + \sqrt{NT})$ for every $\boldsymbol{\sigma} \in NE(COBRA)$. Hence, $\max_{\sigma \in NE(COBRA)} \mathfrak{R}_T(COBRA, \boldsymbol{\sigma}) = \tilde{O}(d\sqrt{T} + \sqrt{NT})$.*

*Proof.* Recall $\mathcal{A}_t$ denotes the set of arms' feature corresponding to the active agents in the round $t$. The regret of COBRA for $\boldsymbol{\sigma} \in$ NE(COBRA) is given as follows:

$$\mathfrak{R}_T(\text{COBRA}, \boldsymbol{\sigma}) = \sum_{t=1}^{T} \left( \theta_\star^\top x_{t,a_t^\star}^\star - \theta_\star^\top x_{t,a_t}^\star \right) = \sum_{t=1}^{T} \left( \max_{a \in \mathcal{A}_t} \theta_\star^\top x_{t,a}^\star - \theta_\star^\top x_{t,a_t}^\star \right). \tag{15}$$

Under Assumption 1, if COBRA selects $a_t \in \mathcal{A}_t : a_t \neq a_t^\star$, we have $\theta_\star^\top x_{t,a_t^\star}^\star \leq \theta_\star^\top x_{t,a_t^\star} \leq \text{UCB}_t(x_{t,a_t^\star}) \leq \text{UCB}_t(x_{t,a_t})$. Using $\theta_\star^\top x_{t,a_t^\star}^\star \leq \text{UCB}_t(x_{t,a_t})$ inequality with Lemma 4, we have

$$\mathfrak{R}_T(\text{COBRA}, \boldsymbol{\sigma}) = \sum_{t=1}^{T} \left( \theta_\star^\top x_{t,a_t^\star}^\star - \theta_\star^\top x_{t,a_t}^\star \right)$$

$$\leq \sum_{t=1}^{T} \left( \theta_\star^\top x_{t,a_t^\star} - \theta_\star^\top x_{t,a_t}^\star \right) \quad \text{(agents are over-reporting)}$$

$$\leq \sum_{t=1}^{T} \left( \text{UCB}_t(x_{t,a_t^\star}) - \theta_\star^\top x_{t,a_t}^\star \right) \quad \text{(first part of Assumption 1)}$$

$$\leq \sum_{t=1}^{T} \left( \text{UCB}_t(x_{t,a_t}) - \theta_\star^\top x_{t,a_t}^\star \right) \quad \text{(as selected arm is } a_t\text{)}$$

$$\leq \sum_{t=1}^{T} \left( \text{UCB}_{t,-a}(x_{t,a_t}) - \theta_\star^\top x_{t,a_t}^\star \right) \quad \text{(second part of Assumption 1)}$$

$$= \sum_{t=1}^{T} \mathbb{1}(a_t = a) \left( \text{UCB}_{t,-a}(x_{t,a}) - \theta_\star^\top x_{t,a}^\star \right)$$

$$= \sum_{a=1}^{N} \sum_{t \leq T, a_t = a} \left( \text{UCB}_{t,-a}(x_{t,a}) - \theta_\star^\top x_{t,a}^\star \right)$$

$$\leq \sum_{a=1}^{N} \left( \sum_{t \leq T, a_t = a} 2\alpha_{t,-a} \|x_{t,a}\|_{V_{t,-a}^{-1}} + 2\sqrt{2R^2 S_t(a) \log(1/\delta_{t,a}^y)} \right) \quad \text{(Lemma 4)}$$

$$= \sum_{a=1}^{N} \sum_{t \leq T, a_t = a} 2\alpha_{t,-a} \|x_{t,a}\|_{V_{t,-a}^{-1}} + \sum_{a=1}^{N} 2\sqrt{2R^2 S_t(a) \log(1/\delta_{t,a}^y)}$$

$$= \sum_{t=1}^{T} 2\alpha_{t,-a} \|x_{t,a_t}\|_{V_{t,-a}^{-1}} + \sum_{a=1}^{N} 2\sqrt{2R^2 S_t(a) \log(1/\delta_{t,a}^y)}. \tag{16}$$

First, we will upper bound the first part of the above inequality, i.e., $\sum_{t=1}^{T} \alpha_{t,-a} \|x_{t,a_t}\|_{V_{t,-a}^{-1}}$, as follows:

$$\sum_{t=1}^{T} 2\alpha_{t,-a} \|x_{t,a_t}\|_{V_{t,-a}^{-1}} \leq 2\sqrt{T} \sqrt{\sum_{t=1}^{T} \left[ \alpha_{t,-a} \|x_{t,a_t}\|_{V_{t,-a}^{-1}} \right]^2}$$

$$\leq 2\sqrt{T}\sqrt{\sum_{t=1}^{T}\left[\alpha_T \left\| x_{t,a_t}\right\|_{V_{t,-a}^{-1}}\right]^2}$$

$$= 2\sqrt{T}\sqrt{\alpha_T^2 \sum_{t=1}^{T}\left\| x_{t,a_t}\right\|_{V_{t,-a}^{-1}}^2}$$

$$= 2\alpha_T\sqrt{T}\sqrt{\sum_{t=1}^{T}\left\| x_{t,a_t}\right\|_{V_{t,-a}^{-1}}^2}$$

$$= 2\alpha_T\sqrt{T}\sqrt{\sum_{t=1}^{T}\left\| x_{t,a_t}\right\|_{V_{t,-a}^{-1}}^2 \frac{\left\| x_{t,a_t}\right\|_{V_t^{-1}}^2}{\left\| x_{t,a_t}\right\|_{V_t^{-1}}^2}}$$

$$= 2\alpha_T\sqrt{T}\sqrt{\sum_{t=1}^{T}\left\| x_{t,a_t}\right\|_{V_t^{-1}}^2 \frac{\left\| x_{t,a_t}\right\|_{V_{t,-a}^{-1}}^2}{\left\| x_{t,a_t}\right\|_{V_t^{-1}}^2}}$$

$$\leq 2\alpha_T C\sqrt{T}\sqrt{\sum_{t=1}^{T}\left\| x_{t,a_t}\right\|_{V_t^{-1}}^2}$$

$$\leq 2\alpha_T C\sqrt{T}\sqrt{2\log\frac{\det(V_T)}{\det(\lambda I_d)}}$$

$$\implies \Re_T\left(\text{COBRA}, \boldsymbol{\sigma}\right) \leq 2\alpha_T C\sqrt{2dT\log(\lambda + TL/d)} = \tilde{O}(d\sqrt{T}). \tag{17}$$

[[Here, using the fact and Lemma the new bound is in terms of $\lambda_{min}$ and $\lambda_{max}$. Then a Corollary should be stated where the regret bound is $O(T^{3/4})$. The similar thing should be mentioned for the non-linear counter part.]]

The first inequality is due to using Cauchy-Schwarz inequality, where the second inequality follows from the fact that $\alpha_{t,-a}$ increases with $t$. The third inequality follows from Lemma 12 of Abbasi-Yadkori et al. (2011), by adapting it to our setting. The fourth inequality follows from the fact that there exists an universal constant $C$ such that $C \geq \max_a \sqrt{\frac{\left\| x_{t,a_t}\right\|_{V_{t,-a}^{-1}}^2}{\left\| x_{t,a_t}\right\|_{V_t^{-1}}^2}}$ for all $t \geq 1$. The last two inequalities follow from Lemma 10 and Lemma 11 of Abbasi-Yadkori et al. (2011), respectively, and $\alpha_T = \tilde{O}(d\log T)$. For first part of Eq. (16), we have $\sum_{t=1}^{T} 2\alpha_{t,-a}\left\| x\right\|_{V_{t,-a}^{-1}} \leq 2\alpha_T\sqrt{2dT(1+C)}\log(\lambda + TL/d)$ from Eq. (17), and then using the Jensen's inequality for the second part with the fact that $\sum_{a=1}^{N} S_t(a) \leq T$. Then, we have

$$\Re_T\left(\text{COBRA}, \boldsymbol{\sigma}\right) \leq 2\alpha_T C\sqrt{2dT\log(\lambda + TL/d)} + 2\sqrt{2R^2 NT\log(1/\delta_{t,a}^y)}$$

$$\implies \Re_T\left(\text{COBRA}, \boldsymbol{\sigma}\right) \leq \tilde{O}(d\sqrt{T} + \sqrt{NT}). \tag{18}$$

We now prove that being truthful is an approximate Nash equilibrium for COBRA. Recall Eq. (10), $S_T(a)$ denotes the number of times an agent being selected by COBRA, which is given as follows:

$$S_T(a) = \sum_{t=1}^{T}\mathbb{1}(a_t = a) \geq S_T^\star(a) - \sum_{t=1}^{T}\mathbb{1}(a_t \neq a_t^\star). \tag{19}$$

To get the lower bound $S_T(a)$, we get the upper bound $\sum_{t=1}^{T}\mathbb{1}(a_t \neq a_t^\star)$ when any agent can behave strategically. Recall $\Delta_{a_t} = \left(\theta_\star^\top x_{t,a_t^\star}^\star - \theta_\star^\top x_{t,a_t}^\star\right) \leq \text{UCB}_{t,-a}(x_{t,a}) - \theta_\star^\top x_{t,a}^\star$ for $a_t \neq a_t^\star$. We multiply and divide $\mathbb{1}(a_t \neq a_t^\star)$ by $\Delta_{a_t}$ as follows:

$$\sum_{t=1}^{T}\mathbb{1}(a_t \neq a_t^\star) = \sum_{t=1}^{T}\mathbb{1}(a_t \neq a_t^\star)\frac{\Delta_{a_t}}{\Delta_{a_t}}$$

$$= \sum_{k=1}^{N} \sum_{t \leq T, a_t = a} \mathbb{1}(a_t \neq a_t^\star, a_t = a) \frac{\Delta_{a_t}}{\Delta_{a_t}}$$

$$\leq \sum_{k=1}^{N} \sum_{t \leq T, a_t = a} \mathbb{1}(a_t \neq a_t^\star, a_t = a) \frac{\text{UCB}_{t,-a}(x_{t,a}) - \theta_\star^\top x_{t,a}^\star}{\Delta_{a_t}}$$

$$\leq \sum_{k=1}^{N} \sum_{t \leq T, a_t = a} \frac{\text{UCB}_{t,-a} - \theta_\star^\top x_{t,a}^\star}{\Delta_{a_t}}$$

Assuming there exists a $\Delta_{\min}$ such that $\Delta_{\min} = \min_{a_t \neq a_t^\star} \Delta_{a_t}$, we get

$$\leq \frac{1}{\Delta_{\min}} \sum_{k=1}^{N} \sum_{t \leq T, a_t = a} \text{UCB}_{t,-a}(x_{t,a}) - \theta_\star^\top x_{t,a}^\star$$

Using Eq. (16) with its upper bound, we have

$$\sum_{t=1}^{T} \mathbb{1}(a_t \neq a_t^\star) \leq \frac{2}{\Delta_{\min}} \left( 2\alpha_T C \sqrt{2dT \log(\lambda + TL/d)} + 2\sqrt{2R^2 NT \log(1/\delta_{t,a}^y)} \right)$$

$$\implies \sum_{t=1}^{T} \mathbb{1}(a_t \neq a_t^\star) \leq \tilde{O}\left( d\sqrt{T} + \sqrt{NT} \right). \tag{20}$$

Using this upper bound in Eq. (19), we get the following bound for any agent $a \in \mathcal{A}$:

$$S_T(a) \geq S_T^\star(a) - \tilde{O}\left( d\sqrt{T} + \sqrt{NT} \right). \tag{21}$$

Now, we consider the case where an agent $a$ deviates from the truthful strategy. Recall Eq. (13), the number of times an agent being selected by COBRA is given as follows:

$$S_T(a) = \sum_{t=1}^{T} \mathbb{1}(a_t = a) \leq \sum_{t=1}^{T} \mathbb{1}(a_t^\star = a) + \sum_{t=1}^{T} \mathbb{1}(a_t \neq a_t^\star). \tag{22}$$

Using Eq. (20) in Eq. (22), we get

$$S_T(a) \leq S_T^\star(a) + \tilde{O}\left( d\sqrt{T} + \sqrt{NT} \right). \tag{23}$$

Combining Eq. (21) and Eq. (23) completes our proof that COBRA is $\tilde{O}\left( d\sqrt{T} + \sqrt{NT} \right)$-NE. $\square$

## C  NON-LINEAR REWARD FUNCTION

Table 1: Examples of different $h(x, \mathcal{O}_t)$ values for some LOOM-compatible contextual bandit algorithms, using notations from the original papers.

| Contextual bandit algorithm | $h(x, \mathcal{O}_t)$ |
|---|---|
| Lin-UCB (Chu et al., 2011) | $\left( R\sqrt{d \log\left( \frac{1 + \frac{tL^2}{\lambda}}{\delta} \right)} + \lambda^{\frac{1}{2}} S \right) \|x\|_{V_t^{-1}}$ |
| GLM-UCB (Li et al., 2017) | $\sqrt{\frac{d}{2} \log(1 + 2t/d) + \log(1/\delta)} \frac{\|x\|_{V_t^{-1}}}{\kappa}$ |
| IGP-UCB (Chowdhury and Gopalan, 2017) | $\left( \sqrt{2(\gamma_{t-1} + 1 + \log(1/\delta))} + B \right) \sigma_{t-1}(x)$ |

### C.1  THEORETICAL RESULTS

We first derive results similar to Lemma 2 and Lemma 3 for contextual bandit problems with non-linear reward functions. For brevity, we ignore $\mathfrak{A}$ in $f_t^{\mathfrak{A}}$ and use only $f_t$.

**Lemma 5.** *Let $\mathfrak{A}$ be a LOOM-compatible contextual bandit algorithm for which $|f_t(x) - f(x)| \leq h(x, \mathcal{O}_t)$ holds with probability at least $1 - \delta$ for any $x \in \mathcal{X}$. Then, for all $t \geq 1$,*

    *1. With probability at least $1 - \delta$,*

$$\forall x \in \mathcal{X} : f(x) \leq UCB_t(x) = f_t(x) + h(x, \mathcal{O}_t).$$

    *2. Similarly, with probability at least $1 - \delta$,*

$$\forall x \in \mathcal{X} : f(x) \geq LCB_t(x) = f_t(x) - h(x, \mathcal{O}_t).$$

*Proof.* The proofs of these results follow directly from the first part of Definition 4. For completeness, we provide the proof of the first part, which follows from straightforward algebraic simplifications of $|f_t(x) - f(x)| \leq h(x, \mathcal{O}_t)$:

$$
\begin{aligned}
&|f_t(x) - f(x)| \leq h(x, \mathcal{O}_t) \\
&\implies |f(x) - f_t(x)| \leq h(x, \mathcal{O}_t) \\
&\implies f(x) - f_t(x) \leq h(x, \mathcal{O}_t) \\
&\implies f(x) \leq f_t(x) + h(x, \mathcal{O}_t).
\end{aligned}
$$

Similarly, the proof of the second part follows with some simple algebraic simplifications of $|f_t(x) - f(x)| \leq h(x, \mathcal{O}_t)$:

$$
\begin{aligned}
&|f_t(x) - f(x)| \leq h(x, \mathcal{O}_t) \\
&\implies f_t(x) - f(x) \leq h(x, \mathcal{O}_t) \\
&\implies f_t(x) - h(x, \mathcal{O}_t) \leq f(x) \\
&\implies f(x) \geq f_t(x) - h(x, \mathcal{O}_t). \qquad \square
\end{aligned}
$$

**Lemma 6.** *Let $\mathfrak{A}$ be a LOOM-compatible contextual bandit algorithm for which $|f_{t,-a}(x) - f(x)| \leq h(x, \mathcal{O}_{t,-a})$ holds with probability at least $1 - \delta$ for any $x \in \mathcal{X}$. Then, for all $t \geq 1$,*

    *1. With probability at least $1 - \delta$,*

$$\forall x \in \mathcal{X} : f(x) \leq UCB_{t,-a}(x) = f_{t,-a}(x) + h(x, \mathcal{O}_{t,-a}).$$

    *2. Similarly, with probability at least $1 - \delta$,*

$$\forall x \in \mathcal{X} : f(x) \geq LCB_{t,-a}(x) = f_{t,-a}(x) - h(x, \mathcal{O}_{t,-a}).$$

*Proof.* The proofs of these results follow directly from the second part of Definition 4. For completeness, we provide the proof of the first part, which follows from straightforward algebraic simplifications of $|f_t(x) - f(x)| \leq h(x, \mathcal{O}_t)$:

$$
\begin{aligned}
&|f_{t,-a}(x) - f(x)| \leq h(x, \mathcal{O}_{t,-a}) \\
&\implies |f(x) - f_{t,-a}(x)| \leq h(x, \mathcal{O}_{t,-a}) \\
&\implies f(x) - f_{t,-a}(x) \leq h(x, \mathcal{O}_{t,-a}) \\
&\implies f(x) \leq f_{t,-a}(x) + h(x, \mathcal{O}_{t,-a}).
\end{aligned}
$$

Similarly, the proof of the second part follows with some simple algebraic simplifications of $|f_t(x) - f(x)| \leq h(x, \mathcal{O}_t)$:

$$
\begin{aligned}
&|f_{t,-a}(x) - f(x)| \leq h(x, \mathcal{O}_{t,-a}) \\
&\implies f_{t,-a}(x) - f(x) \leq h(x, \mathcal{O}_{t,-a}) \\
&\implies f_{t,-a}(x) - h(x, \mathcal{O}_{t,-a}) \leq f(x) \\
&\implies f(x) \geq f_{t,-a}(x) - h(x, \mathcal{O}_{t,-a}). \qquad \square
\end{aligned}
$$

*Proof.* When all agents report truthfully, our algorithm is the same as contextual bandit algorithm $\mathfrak{A}$ with a mechanism for identifying strategic agents that holds with probability at least $1 - \delta_x - \delta_y$. For completeness, we first recall the definition the regret of COBRA as follows:

$$\mathfrak{R}_T\left(\text{COBRA}(\mathfrak{A}), \sigma^\star\right) = \sum_{t=1}^{T} \left( f(x_{t,a_t^\star}^\star) - f(x_{t,a_t}^\star) \right). \tag{24}$$

Since the true feature vector is the same as the reported context-arm feature vector (i.e., $x_{t,a}^\star = x_{t,a}$), we can start with upper bounding the difference $f(x_{t,a_t^\star}^\star) - f(x_{t,a_t}^\star)$ as follows:

$$
\begin{aligned}
f(x_{t,a_t^\star}^\star) - f(x_{t,a_t}^\star) &\le \text{UCB}(x_{t,a_t^\star}) - f(x_{t,a_t}^\star) \\
&\le \text{UCB}(x_{t,a_t}) - f(x_{t,a_t}^\star) \quad \left(\text{as } \text{UCB}(x_{t,a_t^\star}) \le \text{UCB}(x_{t,a_t})\right) \\
&= f_t(x_{t,a_t}) + h(x_{t,a_t}, \mathcal{O}_t) - f(x_{t,a_t}^\star) \\
&\le |f_t(x_{t,a_t}) - f(x_{t,a_t}^\star)| + h(x_{t,a_t}, \mathcal{O}_t) \\
&\le h(x_{t,a_t}, \mathcal{O}_t) + h(x_{t,a_t}, \mathcal{O}_t) \\
\implies f(x_{t,a_t^\star}^\star) - f(x_{t,a_t}^\star) &\le 2h(x_{t,a_t}, \mathcal{O}_t). \tag{25}
\end{aligned}
$$

Note that $f_t$ is an estimator of the reward function $f$ as the true feature vector is the same as the reported context-arm feature vector. After using the upper bound given in Eq. (25) into Eq. (24), we get an upper bound on the regret as follows:

$$
\begin{aligned}
\mathfrak{R}_T\left(\text{COBRA}(\mathfrak{A}), \sigma^\star\right) &= \sum_{t=1}^{T} \left( f(x_{t,a_t^\star}^\star) - f(x_{t,a_t}^\star) \right) \\
&\le \sum_{t=1}^{T} 2h(x_{t,a_t}, \mathcal{O}_t) \\
&= 2 \sum_{t=1}^{T} h(x_{t,a_t}, \mathcal{O}_t) \\
&\le 2 \sum_{t=1}^{T} \sqrt{T \sum_{t=1}^{T} [h(x_{t,a_t}, \mathcal{O}_t)]^2} \\
\implies \mathfrak{R}_T\left(\text{COBRA}(\mathfrak{A}), \sigma^\star\right) &\le 2\sqrt{T} \sqrt{\sum_{t=1}^{T} [h(x_{t,a_t}, \mathcal{O}_t)]^2} = \tilde{O}\left(\tilde{d}\sqrt{T}\right). \tag{26}
\end{aligned}
$$

The first inequality directly follows from Eq. (25). The second inequality is due to using Cauchy-Schwarz inequality. The last equality is due to $\sqrt{\sum_{t=1}^{T} [h(x_{t,a_t}, \mathcal{O}_t)]^2} = \tilde{O}(\tilde{d} \log T)$.

We now prove that being truthful is an approximate Nash equilibrium for COBRA. Recall, $S_T(a)$ denotes the number of times an agent being selected by COBRA, which is given as follows:

$$
\begin{aligned}
S_T(a) &= \sum_{t=1}^{T} \mathbb{1}(a_t = a) \\
&= \sum_{t=1}^{T} \mathbb{1}(a_t = a, a_t^\star = a) + \sum_{t=1}^{T} \mathbb{1}(a_t = a, a_t^\star \ne a) \\
&\ge \sum_{t=1}^{T} \mathbb{1}(a_t^\star = a) - \sum_{t=1}^{T} \mathbb{1}(a_t^\star = a, a_t \ne a) \\
&\ge \sum_{t=1}^{T} \mathbb{1}(a_t^\star = a) - \sum_{t=1}^{T} \mathbb{1}(a_t \ne a_t^\star)
\end{aligned}
$$

$$\implies S_T(a) \geq S_T^\star(a) - \sum_{t=1}^{T} \mathbb{1}(a_t \neq a_t^\star). \tag{27}$$

To get the lower bound $S_T(a)$, we get an upper bound $\sum_{t=1}^{T} \mathbb{1}(a_t \neq a_t^\star)$. Let $\Delta_{a_t} = \left(f(x_{t,a_t^\star}^\star) - f(x_{t,a_t}^\star)\right) > 0$ for $a_t \neq a_t^\star$. We multiply and divide $\mathbb{1}(a_t \neq a_t^\star)$ by $\Delta_{a_t}$ and then use inequality in Eq. (25), i.e., $\Delta_{a_t} \leq 2h(x_{t,a_t}, \mathcal{O}_t)$ as follows:

$$\sum_{t=1}^{T} \mathbb{1}(a_t \neq a_t^\star) = \sum_{t=1}^{T} \mathbb{1}(a_t \neq a_t^\star) \frac{\Delta_{a_t}}{\Delta_{a_t}}$$

$$\leq \sum_{t=1}^{T} \mathbb{1}(a_t \neq a_t^\star) \frac{2h(x_{t,a_t}, \mathcal{O}_t)}{\Delta_{a_t}} \quad (\text{as } x_{t,a}^\star = x_{t,a})$$

$$\leq \sum_{t=1}^{T} \frac{2h(x_{t,a_t}, \mathcal{O}_t)}{\Delta_{a_t}}$$

$$= \sum_{t=1}^{T} \frac{1}{\Delta_{a_t}} 2h(x_{t,a_t}, \mathcal{O}_t)$$

$$\leq \sum_{t=1}^{T} \frac{1}{\Delta_{\min}} 2h(x_{t,a_t}, \mathcal{O}_t)$$

$$= \frac{2}{\Delta_{\min}} \sum_{t=1}^{T} h(x_{t,a_t}, \mathcal{O}_t)$$

$$\leq \frac{2}{\Delta_{\min}} \sqrt{T} \sqrt{\sum_{t=1}^{T} [h(x_{t,a_t}, \mathcal{O}_t)]^2}$$

$$\implies \sum_{t=1}^{T} \mathbb{1}(a_t \neq a_t^\star) \leq \tilde{O}\left(\tilde{d}\sqrt{T}\right). \tag{28}$$

Note that $\Delta_{\min} = \min_{a_t \neq a_t^\star} \Delta_{a_t}$. Although using $\Delta_{\min}$ loosen the upper bound, we use this to get dependence on $T$. Using this upper bound in Eq. (27), we get the following bound for any agent $a \in \mathcal{A}$:

$$S_T(a) \geq S_T^\star(a) - \tilde{O}\left(\tilde{d}\sqrt{T}\right). \tag{29}$$

Now we consider the case where an agent $a$ deviates from the truthful strategy. The number of times an agent being selected by COBRA is given as follows:

$$S_T(a) = \sum_{t=1}^{T} \mathbb{1}(a_t = a)$$

$$= \sum_{t=1}^{T} \mathbb{1}(a_t = a, a_t^\star = a) + \sum_{t=1}^{T} \mathbb{1}(a_t = a, a_t^\star \neq a)$$

$$\leq \sum_{t=1}^{T} \mathbb{1}(a_t^\star = a) + \sum_{t=1}^{T} \mathbb{1}(a_t = a, a_t^\star \neq a)$$

$$\leq \sum_{t=1}^{T} \mathbb{1}(a_t^\star = a) + \sum_{t=1}^{T} \mathbb{1}(a_t \neq a_t^\star). \tag{30}$$

Using Eq. (28) in Eq. (30), we get

$$S_T(a) \leq S_T^\star(a) + \tilde{O}\left(\tilde{d}\sqrt{T}\right). \tag{31}$$

Combining Eq. (29) and Eq. (31) completes our proof that COBRA is $\tilde{O}(\tilde{d}\sqrt{T})$-NE. $\qquad\square$

We need the following result that upper bound the total amount of regret that an agent $a$ can exert before being identified by LOOM.

**Lemma 7.** *Let* $UCB_{t,-a}(x_{s,a}) = f_{t,-a}(x_{s,a}) + h(x_{s,a}, \mathcal{O}_{t,-a})$, *where* $x_{s,a}$ *is the arm feature vector associated with agent* $a$ *in the round* $s$. *Then, with probability at least* $1 - \delta_{t,a}^x - \delta_{t,a}^y$,

$$\sum_{s \leq t:\, a_s = a} \left( UCB_{t,-a}(x_{s,a}) - f(x_{s,a}^\star) \right) \leq \sum_{s \leq t, a_s = a} 2h(x_{s,a}, \mathcal{O}_{t,-a}) + 2\sqrt{2R^2 S_t(a) \log(1/\delta_{t,a}^y)}.$$

*Proof.* Using Eq. (6) with Lemma 3 for non-linear reward function that holds with probability at least $1 - \delta_{t,a}^x$, we get:

$$\sum_{s \leq t, a_s = a} (f_{t,-a}(x_{s,a}) + h(x_{s,a}, \mathcal{O}_{t,-a})) \leq \sum_{s \leq t, a_s = a} y_s + \sqrt{2R^2 S_t(a) \log(1/\delta_{t,a}^y)}$$

$$\leq \sum_{s \leq t, a_s = a} f(x_{s,a}^\star) + \sqrt{2R^2 S_t(a) \log(1/\delta_{t,a}^y)}$$

$$+ \sqrt{2R^2 S_t(a) \log(1/\delta_{t,a}^y)}$$

$$\implies \sum_{s \leq t, a_s = a} \left( f_{t,-a}(x_{s,a}) - f(x_{s,a}^\star) \right) \leq \sum_{s \leq t, a_s = a} h(x_{s,a}, \mathcal{O}_{t,-a}) + 2\sqrt{2R^2 S_t(a) \log(1/\delta_{t,a}^y)}.$$

The second inequality follows from Eq. (4) by using upper bound on $\sum_{s \leq t, a_s = a} y_s$ that holds with probability $1 - \delta_{t,a}^y$. Now we prove the second part of the result by replacing $f_{t,-a}(x_{s,a})$ by $UCB_{t,-a}(x_{s,a}) - h(x_{s,a}, \mathcal{O}_{t,-a})$ and we get

$$\sum_{s \leq t, a_s = a} \left( UCB_{t,-a}(x_{s,a}) - h(x, \mathcal{O}_{t,-a}) - f(x_{s,a_s}^\star) \right)$$

$$\leq \sum_{s \leq t, a_s = a} h(x_{s,a}, \mathcal{O}_{t,-a}) + 2\sqrt{2R^2 S_t(a) \log(1/\delta_{t,a}^y)}$$

$$\implies \sum_{s \leq t, a_s = a} \left( UCB_{t,-a}(x_{s,a}) - f(x_{s,a_s}^\star) \right)$$

$$\leq \sum_{s \leq t, a_s = a} 2h(x_{s,a}, \mathcal{O}_{t,-a}) + 2\sqrt{2R^2 S_t(a) \log(1/\delta_{t,a}^y)}. \qquad \square$$

**Theorem 5** (Non-linear). *Let* $\mathfrak{A}$ *be a LOOM-compatible contextual bandit algorithm used by COBRA. If Assumption 1 hold then, the regret of COBRA is* $\mathfrak{R}_T(COBRA(\mathfrak{A}), \boldsymbol{\sigma}) = \tilde{O}(\tilde{d}\sqrt{T} + \sqrt{NT})$ *for every* $\boldsymbol{\sigma} \in NE(COBRA(\mathfrak{A}))$. *Hence,* $\max_{\sigma \in NE(COBRA(\mathfrak{A}))} \mathfrak{R}_T(COBRA(\mathfrak{A}), \boldsymbol{\sigma}) = \tilde{O}(\tilde{d}\sqrt{T} + \sqrt{NT})$.

*Proof.* Recall $\mathcal{A}_t$ denotes the set of arms' feature corresponding to the active agents in the round $t$. The regret of COBRA for $\boldsymbol{\sigma} \in NE(COBRA(\mathfrak{A}))$ is given as follows:

$$\mathfrak{R}_T(COBRA(\mathfrak{A}), \boldsymbol{\sigma}) = \sum_{t=1}^T \left( f(x_{t,a_t^\star}^\star) - f(x_{t,a_t}^\star) \right) = \sum_{t=1}^T \left( \max_{a \in \mathcal{A}_t} f(x_{t,a}^\star) - f(x_{t,a_t}^\star) \right). \quad (32)$$

Under Assumption 1, if COBRA selects $a_t \in \mathcal{A}_t : a_t \neq a_t^\star$, we have $f(x_{t,a_t^\star}^\star) \leq f(x_{t,a_t^\star}) \leq UCB_t(x_{t,a_t^\star}) \leq UCB_t(x_{t,a_t})$. Using $f(x_{t,a_t^\star}^\star) \leq UCB_t(x_{t,a_t})$ inequality with Lemma 7, we have

$$\mathfrak{R}_T(COBRA(\mathfrak{A}), \boldsymbol{\sigma}) = \sum_{t=1}^T \left( f(x_{t,a_t^\star}^\star) - f(x_{t,a_t}^\star) \right)$$

$$\leq \sum_{t=1}^T \left( f(x_{t,a_t^\star}) - f(x_{t,a_t}^\star) \right) \quad \text{(agents are over-reporting)}$$

$$\leq \sum_{t=1}^T \left( UCB_t(x_{t,a_t^\star}) - f(x_{t,a_t}^\star) \right) \quad \text{(first part of Assumption 1)}$$

$$\leq \sum_{t=1}^{T} \left( \text{UCB}_t(x_{t,a_t}) - f(x_{t,a_t}^\star) \right) \qquad \text{(as selected arm is } a_t)$$

$$\leq \sum_{t=1}^{T} \left( \text{UCB}_{t,-a}(x_{t,a_t}) - f(x_{t,a_t}^\star) \right) \qquad \text{(second part of Assumption 1)}$$

$$= \sum_{t=1}^{T} \mathbb{1}(a_t = a) \left( \text{UCB}_{t,-a}(x_{t,a}) - f(x_{t,a}^\star) \right)$$

$$= \sum_{a=1}^{N} \sum_{t \leq T, a_t = a} \left( \text{UCB}_{t,-a}(x_{t,a}) - f(x_{t,a}^\star) \right)$$

$$\leq \sum_{a=1}^{N} \left( \sum_{t \leq T, a_t = a} 2h(x_{t,a}, \mathcal{O}_{t,-a}) + 2\sqrt{2R^2 S_t(a) \log(1/\delta_{t,a}^y)} \right)$$

$$= \sum_{a=1}^{N} \sum_{t \leq T, a_t = a} 2h(x_{t,a}, \mathcal{O}_{t,-a}) + \sum_{a=1}^{N} 2\sqrt{2R^2 S_t(a) \log(1/\delta_{t,a}^y)}$$

$$= \sum_{t=1}^{T} 2h(x_{t,a_t}, \mathcal{O}_{t,-a}) + \sum_{a=1}^{N} 2\sqrt{2R^2 S_t(a) \log(1/\delta_{t,a}^y)}$$

$$\leq 2\sqrt{T} \sqrt{\sum_{t=1}^{T} [h(x_{t,a_t}, \mathcal{O}_{t,-a})]^2} + 2 \sum_{a=1}^{N} \sqrt{2R^2 S_t(a) \log(1/\delta_{t,a}^y)}$$

$$\leq 2\sqrt{T} \sqrt{\sum_{t=1}^{T} [h(x_{t,a_t}, \mathcal{O}_{t,-a})]^2} + 2\sqrt{2R^2 NT \log(1/\delta_{t,a}^y)}$$

$$\leq 2C\sqrt{T} \sqrt{\sum_{t=1}^{T} [h(x_{t,a_t}, \mathcal{O}_{t,a})]^2} + 2\sqrt{2R^2 NT \log(1/\delta_{t,a}^y)}$$

$$\implies \mathfrak{R}_T(\text{COBRA}(\mathfrak{A}), \boldsymbol{\sigma}) = \tilde{O}\left( \tilde{d}\sqrt{T} + \sqrt{NT} \right). \tag{33}$$

The third-last inequality follows from Lemma 7. The second-last inequality is due to using Cauchy-Schwarz inequality where last inequality follows from Jensen's inequality with the fact that $\sum_{a=1}^{N} S_t(a) \leq T$.

We now prove that being truthful is an approximate Nash equilibrium for COBRA. Recall Eq. (27), $S_T(a)$ denotes the number of times an agent being selected by COBRA, which is given as follows:

$$S_T(a) = \sum_{t=1}^{T} \mathbb{1}(a_t = a) \geq S_T^\star(a) - \sum_{t=1}^{T} \mathbb{1}(a_t \neq a_t^\star). \tag{34}$$

To get the lower bound $S_T(a)$, we get the upper bound $\sum_{t=1}^{T} \mathbb{1}(a_t \neq a_t^\star)$ when any agent can behave strategically. Recall $\Delta_{a_t} = \left( f(x_{t,a_t^\star}^\star) - f(x_{t,a_t}^\star) \right) \leq \text{UCB}_{t,-a}(x_{t,a}) - f(x_{t,a}^\star)$ for $a_t \neq a_t^\star$. We multiply and divide $\mathbb{1}(a_t \neq a_t^\star)$ by $\Delta_{a_t}$ as follows:

$$\sum_{t=1}^{T} \mathbb{1}(a_t \neq a_t^\star) = \sum_{t=1}^{T} \mathbb{1}(a_t \neq a_t^\star) \frac{\Delta_{a_t}}{\Delta_{a_t}}$$

$$= \sum_{k=1}^{N} \sum_{t \leq T, a_t = a} \mathbb{1}(a_t \neq a_t^\star, a_t = a) \frac{\Delta_{a_t}}{\Delta_{a_t}}$$

$$\leq \sum_{k=1}^{N} \sum_{t \leq T, a_t = a} \mathbb{1}(a_t \neq a_t^\star, a_t = a) \frac{\text{UCB}_{t,-a}(x_{t,a}) - f(x_{t,a}^\star)}{\Delta_{a_t}}$$

$$\leq \sum_{k=1}^{N} \sum_{t \leq T, a_t = a} \frac{\text{UCB}_{t,-a} - f(x_{t,a}^{\star})}{\Delta_{a_t}}$$

Assuming there exists a $\Delta_{\min}$ such that $\Delta_{\min} = \min_{a_t \neq a_t^{\star}} \Delta_{a_t}$, we get

$$\leq \frac{1}{\Delta_{\min}} \sum_{k=1}^{N} \sum_{t \leq T, a_t = a} \text{UCB}_{t,-a}(x_{t,a}) - f(x_{t,a}^{\star})$$

Using Eq. (33) with its upper bound, we have

$$\sum_{t=1}^{T} \mathbb{1}(a_t \neq a_t^{\star}) \leq \frac{2}{\Delta_{\min}} \left( 2\sqrt{T} \sqrt{\sum_{t=1}^{T} [h(x_{t,a_t}, \mathcal{O}_{t,-a})]^2} + 2\sqrt{2R^2 NT \log(1/\delta_{t,a}^{y})} \right) \quad (35)$$

$$\implies \sum_{t=1}^{T} \mathbb{1}(a_t \neq a_t^{\star}) = \tilde{O}\left( d\sqrt{T} + \sqrt{NT} \right).$$

Using this upper bound in Eq. (34), we get the following bound for any agent $a \in \mathcal{A}$:

$$S_T(a) \geq S_T^{\star}(a) - \tilde{O}\left( \tilde{d}\sqrt{T} + \sqrt{NT} \right). \quad (36)$$

Now, we consider the case where an agent $a$ deviates from the truthful strategy. Recall Eq. (30), the number of times an agent being selected by COBRA is given as follows:

$$S_T(a) = \sum_{t=1}^{T} \mathbb{1}(a_t = a) \leq \sum_{t=1}^{T} \mathbb{1}(a_t^{\star} = a) + \sum_{t=1}^{T} \mathbb{1}(a_t \neq a_t^{\star}). \quad (37)$$

Using Eq. (35) in Eq. (37), we get

$$S_T(a) \leq S_T^{\star}(a) + \tilde{O}\left( \tilde{d}\sqrt{T} + \sqrt{NT} \right). \quad (38)$$

Combining Eq. (36) and Eq. (38) completes our proof that COBRA is $\tilde{O}\left( \tilde{d}\sqrt{T} + \sqrt{NT} \right)$-NE. $\square$

## D   DISCUSSION ABOUT LOOM AND ASSUMPTION 1

### D.1   LOOM-RELATED DISCUSSION

**Example showing impact of arm feature distribution on LOOM.**   To illustrate how arm feature distribution of agents plays out in our setting, consider the example of an online e-commerce platform recommending sellers. When multiple sellers offer similar products, such as round-neck T-shirts priced between \$5 and \$15, their corresponding arms (i.e., T-shirts) will have similar feature vectors. In contrast, sellers offering distinct products, such as one selling T-shirts and another selling smartphones, will likely have arms with very different feature representations. Suppose a seller offers a unique product, for example, being the only seller of Apple products on the online e-commerce platform. In that case, there is no incentive to misreport their features, as no competitors exist. Thus, misreporting becomes strategically beneficial only when sellers offer similar products, in which case their arm features are drawn from similar distributions, allowing LOOM to identify the misreporting agent.

**Failure case of LOOM.**   The estimators used in LOOM cannot estimate directions orthogonal to the available observations, e.g., estimating the mean reward of an arm using others in stochastic $K$-armed bandits. However, these estimators are used solely to identify strategic agents and do not affect the arm-selection strategy of the underlying contextual bandit algorithm. As a result, if a over-reporting agent's arm features occupy distinct regions of the feature space, LOOM may fail to identify the agent, but the arm-selection strategy may remain unaffected due to the non-overlapping feature space.

**Alternative to complete removal of over-reporting agent.**   When an agent over-reports its arm features and is completely eliminated, such an event would be extremely rare in practice. If every

agent were to persistently misreport, any mechanism prioritizing user experience would eventually stop recommending such agents, thereby driving the platform's reward to zero. This outcome is not a flaw but rather a *safeguard preserving user trust*: continuing to recommend only strategic, untrustworthy agents would ultimately diminish both reward and user engagement. An alternative mechanism is that when an agent systematically over-reports, COBRA flags and temporarily removes it. This mechanism discourages agents from gaming the system while restoring recommendation quality and benefiting users and the platform. However, deriving theoretical guarantees for such a mechanism may be non-trivial.

**Agents with multiple arms.** Our results generalize to the setting where each agent controls multiple arms. LOOM maintains individual records of each agent's reported arm feature vectors and reward history for their respective arms, hence applying LOOM's statistical test independently to each agent is possible. This allows LOOM to identify over-reporting agents whose optimistic/pessimistic reward gaps exceed the threshold, even if the agent over-reports its different arms in distinct ways. As long as Assumption 1 holds for each active agent, the regret and approximate equilibrium guarantees continue to apply. Specifically, the regret bound remains $\tilde{O}(d\sqrt{T} + \sqrt{NT})$, where $N$ denotes the number of agents. For very large number of agents ($N$), it may increase computational complexity and affect regret bounds due to the $\sqrt{N}$ term. Note that we can perform all the agents-related computations in parallel as they are independent of each other.

**Sub-optimal agent.** We highlight that, for agents with suboptimal arms, truthful reporting and over-reporting may lead to similar outcomes: either being ignored by the learner or being eliminated. To address this challenge, a promising direction for future work is to incorporate fairness constraints into the arm selection, thereby ensuring that even suboptimal agents have a chance of being selected.

### D.2 Discussion about Assumption 1

We believe that addressing strategic behavior in a contextual bandit setting without relying on monetary incentives is a challenging and underexplored problem. There is limited prior work in the literature on this topic, despite its many practical applications, e.g., online platforms where sellers may attempt to manipulate the contextual information of their products to gain an advantage. Our key contribution is the development of an approximately incentive-compatible property inspired by the VCG mechanism. However, when an agent over-report arm features, all estimators used by LOOM become biased due to the over-reported arm feature vectors as inputs. It happens because the misreported features distort the overall feature distribution, creating mismatches between features and their corresponding rewards, which in turn induces bias in the estimators.

In contextual bandits, estimators (such as those for the reward function parameters) rely on the assumption that observed features and rewards are generated according to an honest, stationary process. When agents systematically over-report by deliberately inflating feature values to increase their selection probability, the samples collected by the algorithm are corrupted: the feature vectors in the data do not match the ground truth. Our next result demonstrates that over-reported arm features increase the bias in estimators used by LOOM that take them as input.

**Lemma 8** (Biased-ness due over-reporting.)**.** *Using over-reported arm features increase the bias in all estimators used by LOOM that take these arm features as input.*

*Proof.* Without loss of generality, consider a linear reward function with an unknown true parameter vector $\theta_\star \in \mathbb{R}^d$, where $d \geq 1$ is the dimension of the context-arm feature vector. At each round $t$, the true feature vector associated with agent $a$ is denoted by $x_{t,a}^\star$. However, the agent may strategically misreport their features as $x_{t,a}$ in an attempt to appear more favorable, i.e., that is, to give the impression of a higher expected reward, satisfying $\theta_\star^\top x_{t,a} \geq \theta_\star^\top x_{t,a}^\star$ Let $\eta_s = \theta_\star^\top x_{s,a_s} - \theta_\star^\top x_{s,a_s}^\star \geq 0$ denote the difference in reward between the misreported features and the true features, where $a_s$ is the agent selected in round $s$. Note that $\eta_s = 0$ when the agent reports truthfully. The ridge regression estimator based on the observed data $\mathcal{H}_t = \{(x_{s,a_s}, y_s)\}_{s<t}$ is given by

$$\hat{\theta}_t = \left(\lambda I + \sum_{s<t} x_{s,a_s} x_{s,a_s}^\top\right)^{-1} \left(\sum_{s<t} x_{s,a_s} y_s\right),$$

where $\lambda > 0$ is the regularization parameter that ensures the matrix $\sum_{s<t} x_{s,a_s} x_{s,a_s}^\top$ is invertible.

Since $y_s = \theta_\star^\top x_{s,a_s}^\star + \varepsilon_s$, which depends only on the true context-arm feature vector, we can rewrite it as $y_s = \theta_\star^\top x_{s,a_s} - \eta_s + \varepsilon_s$. Substituting this into the ridge regression estimator, we obtain:

$$\hat{\theta}_t = \left(\lambda I + \sum_{s<t} x_{s,a_s} x_{s,a_s}^\top\right)^{-1} \left(\sum_{s<t} x_{s,a_s} \left(\theta_\star^\top x_{s,a_s} - \eta_s + \varepsilon_s\right)\right).$$

Let $y_s' = \theta_\star^\top x_{s,a_s} + \varepsilon_s$ denote the noisy reward for the (possibly misreported) context-arm feature vector $x_{s,a_s}$, generated by the true reward function. Using this notation, we can rewrite the estimator as:

$$\hat{\theta}_t = \left(\lambda I + \sum_{s<t} x_{s,a_s} x_{s,a_s}^\top\right)^{-1} \left(\sum_{s<t} x_{s,a_s} (y_s' - \eta_s)\right).$$

Expanding the expression, we obtain:

$$\hat{\theta}_t = \left(\lambda I + \sum_{s<t} x_{s,a_s} x_{s,a_s}^\top\right)^{-1} \left(\sum_{s<t} x_{s,a_s} y_s'\right) - \left(\lambda I + \sum_{s<t} x_{s,a_s} x_{s,a_s}^\top\right)^{-1} \left(\sum_{s<t} x_{s,a_s} \eta_s\right).$$

Let $\tilde{\theta}_t$ denote the ridge regression estimator computed using the misreported feature vectors $\{x_{s,\cdot}\}_{s<t}$ and the corresponding noisy rewards $\{y_s'\}_{s<t}$, i.e.,

$$\tilde{\theta}_t = \left(\lambda I + \sum_{s<t} x_{s,a_s} x_{s,a_s}^\top\right)^{-1} \left(\sum_{s<t} x_{s,a_s} y_s'\right).$$

Then, we can express $\hat{\theta}_t$ as:

$$\hat{\theta}_t = \tilde{\theta}_t - \left(\lambda I + \sum_{s<t} x_{s,a_s} x_{s,a_s}^\top\right)^{-1} \sum_{s<t} \left(x_{s,a_s} \eta_s\right),$$

where the term $\left(\lambda I + \sum_{s<t} x_{s,a_s} x_{s,a_s}^\top\right)^{-1} \sum_{s<t} \left(x_{s,a_s} \eta_s\right)$ represents the additional bias introduced due to the misreporting of features. $\qquad \square$

**Some special cases of Assumption 1.** The biased-ness due to over-reported arm features makes it impossible to derive theoretical guarantees without introducing additional assumptions, such as Assumption 1. To validate its practicality, we consider the following three cases:

**Case 1.** All agents report truthfully: When reported features are the same as true features, i.e., $x = x^*$ for all $x \in \mathcal{X}$, $\text{UCB}_t(x)$ is an upper bound of $\theta_*^\top x$ with probability at least $1 - \delta$ (or with high probability, Lemma 2). As a result, first part of Assumption 1 holds, which is only needed to prove our Theorem 2, and hence the NE and regret bounds of our proposed algorithm, COBRA, are improved by a factor of $\sqrt{N}$ compared to Theorem 5.1 in Kleine Buening et al. (2024).

**Case 2.** One agent can over-report while other agents report truthfully and linear reward function: In Lemma 2, $\alpha_t$ is a non-decreasing function of $t$ that grows logarithmically, while $||x||_{V_t^{-1}}$ converges at a rate of $1/\sqrt{t}$, leading to tighter confidence ellipsoid as $t$ increases. Thus, $\text{UCB}_t(x)$ is smaller than $\text{UCB}_{t,-a}(x)$ for any $x$ due to the use of additional observations from agent $a$. However, when an agent $a$ over-report its features, it leads to biased estimates of $\theta_\star$. Since the agent over-reports, $\hat{\theta}_t$ becomes a downward-biased estimator of $\theta_\star$ (Example 4.7 in Wooldridge, in which over-reporting features can be treated as under-reporting rewards[3]). As a downward biased estimator leads to under-estimation with the fact that $\alpha_t ||x||_{V_t^{-1}}$ is smaller than $\alpha_{t,-a} ||x||_{V_{t,-a}^{-1}}$, $\text{UCB}_t(x)$ is smaller than $\text{UCB}_{t,-a}(x)$ with high probability. If an agent keeps over-reporting, our proposed method, LOOM, will detect this behavior and remove the agent from the active selection pool. Consequently, Assumption 1 first part will hold as the remaining agents are truthful.

**Case 3.** Multiple agents can over-report: In this case, all estimators used by COBRA become biased, making it impossible to derive theoretical guarantees without additional constraints. Our Theorem 4

---

[3]Wooldridge, J. M. (2010). Econometric analysis of cross section and panel data. MIT press.

and Theorem 5 hold as long as Assumption 1 is satisfied. Notably, we impose no restrictions on how agents report their features, aside from no collusion assumption, which is a common assumption in VCG-type mechanisms (Vickrey, 1961; Clarke, 1971; Groves, 1973).

Relaxing this assumption remains an interesting direction for future work.

**Comparison from existing literature.** Kleine Buening et al. (2024) use agent-specific estimators that detect over-reporting in linear contextual bandits. In contrast, our method takes inspiration from the VCG mechanism (Vickrey, 1961; Clarke, 1971; Groves, 1973) and uses the observations associated with other agents to identify the over-reporting of an agent. This key difference leads us to use $\text{LCB}_{t,-a}(x_{t,a})$ (pessimistic reward estimate using observations of all agents except agent $a$) while Kleine Buening et al. (2024) use $\text{LCB}_{t,a}(x_{t,a})$ (pessimistic reward estimate only using observation associated with agent $a$) for detection.

Theorem 5.2 of Kleine Buening et al. (2024) holds under their Assumption 2 (holds only for linear reward functions), which has the following consequence:

$$f(x^*_{t,a^*}) \le \text{UCB}_{t,a^*_t}(x_{t,a^*}) \le \text{UCB}_{t,a_t}(x_{t,a_t})$$

(see proof of Lemma E.5 on Page 27 in Kleine Buening et al. (2024)).

In contrast, our assumptions imply the following:

$$f(x^*_{t,a^*}) \le f(x_{t,a^*}) \le \text{UCB}_t(x_{t,a^*_t}) \le \text{UCB}_t(x_{t,a_t}) \le \text{UCB}_{t,-a}(x_{t,a_t})$$

which gives:

$$f(x^*_{t,a^*}) \le \text{UCB}_t(x_{t,a^*_t}) \le \text{UCB}_{t,-a_t}(x_{t,a_t}) \implies f(x^*_{t,a^*}) \le \text{UCB}_{t,-a_t}(x_{t,a_t}). \tag{39}$$

Assumption 2 of (Kleine Buening et al., 2024) and our Assumption 1 share a key similarity: they define the conditions under which some theoretical results hold (their Theorem 5.2 and ours Theorem 4). These assumptions also lead to similar consequences, i.e., the maximum expected reward in any round is upper-bounded by the optimistic reward estimate of the selected arm (or agent) computed using the same agent(s) (i.e., $\text{UCB}_{t,a_t}(x_{t,a})$ and $\text{UCB}_{t,-a_t}(x_{t,a})$) as used in the mechanism for identifying over-reporting agents. We emphasize that our Assumption 1 is not directly comparable to that of Kleine Buening et al. (2024), as they provide conditions for the theoretical guarantees of algorithms based on different underlying mechanisms.

We would like to highlight that detecting over-reporting using only an agent's own observations may be ineffective in practice, particularly when the true parameter $\theta_*$ is unknown due to the absence of any external baseline for comparison. In contrast, our VCG-inspired approach leverages observations from other agents to identify over-reporting, making it more practical, as the targeted agent cannot directly influence the detection mechanism.

Furthermore, *we extend our analysis to a class of non-linear contextual bandit algorithms, where the confidence ellipsoid around the unknown parameter $\theta_\star$ satisfies certain assumptions, and LOOM can be used as a subroutine in linear contextual bandit algorithms to identify strategic agents.* This constitutes a new contribution within this setting.

**Equivalence between Assumption 1 and Leave-one-out arm selection strategy.** Assumption 1 is equivalent to replacing the arm selection strategy in Step 5 of our proposed algorithm, COBRA, with a leave-one-out (LOO) arm selection strategy defined as:

$$a_t = \arg\max_{a \in \mathcal{A}} \text{UCB}_{t,-a}(x_{t,a}),$$

where $\text{UCB}_{t,-a}(x_{t,a})$ denotes the upper confidence bound computed using a LOO estimator that excludes the historical data from agent $a$, i.e., while calculating the UCBvalue for agent $a$, we use only the data from other agents. However, the confidence bounds derived from these LOO estimators are generally looser than those obtained using the data from all agents. We next prove the equivalence between Assumption 1 and the use of the LOO arm selection strategy.

**Lemma 9.** *When one of the agent over-report, having Assumption 1 results same the regret as using LOO arm selection strategy.*

*Proof.* For completeness, we first derive the consequences of over-reporting under Assumption 1 (also mentioned at the top of Page 20 in the Appendix) as follows. When an agent over-reports, the

following inequality holds for the optimal arm $x_{t,a_t^\star}$ in round $t$: $f(x_{t,a_t^\star}^\star) \leq f(x_{t,a_t^\star})$. Using part 1 of Assumption 1, i.e., $f(x_{t,a_t^\star}) \leq \mathrm{UCB}_t(x_{t,a_t^\star})$, which states that the upper confidence bounds remain valid even when agents over-report, we obtain:

$$f(x_{t,a_t^\star}^\star) \leq f(x_{t,a_t^\star}) \leq \mathrm{UCB}_t(x_{t,a_t^\star}).$$

Since COBRA selects arm $a_t$ over $a_t^\star$, i.e., $\mathrm{UCB}_t(x_{t,a_t^\star}) \leq \mathrm{UCB}_t(x_{t,a_t})$, we obtain:

$$f(x_{t,a_t^\star}^\star) \leq \mathrm{UCB}_t(x_{t,a_t}).$$

Using Part 2 of Assumption 1, i.e., $\mathrm{UCB}_t(x_{t,a_t}) \leq \mathrm{UCB}_{t,-a_t}(x_{t,a_t})$, we get:

$$f(x_{t,a_t^\star}^\star) \leq \mathrm{UCB}_{t,-a_t}(x_{t,a_t}). \tag{40}$$

When we use the LOO arm selection strategy $a_t = \arg\max_{a \in \mathcal{A}} \mathrm{UCB}_{t,-a}(x_{t,a})$, the following chain of inequalities holds under over-reporting:

$$f(x_{t,a_t^\star}^\star) \leq f(x_{t,a_t^\star} \leq \mathrm{UCB}_{t,-a_t^\star}(x_{t,a_t^\star}).$$

Since arm $a_t$ is selected, i.e., $\mathrm{UCB}_{t,-a_t^\star}(x_{t,a_t^\star}) \leq \mathrm{UCB}_{t,-a_t}(x_{t,a_t})$, we conclude:

$$f(x_{t,a_t^\star}^\star) \leq \mathrm{UCB}_{t,-a_t}(x_{t,a_t}). \tag{41}$$

Note that Eq. (40) and Eq. (41) are equivalent, Assumption 1 and the leave-one-out (LOO) arm selection strategy lead to the same theoretical consequences, hence having same NE and regret guarantees. $\square$

To obtain an upper bound on the regret, we derive a key result that bounds the difference $\mathrm{UCB}_{t,-a_t}(x_{t,a_t}) - f(x_{t,a_t^\star})$. These results are formalized in Lemma 4 for the linear reward function and in Lemma 7 for the non-linear case. Importantly, our mechanism, LOOM, ensures that truthful reporting is a dominant strategy for each agent when all others report truthfully. Therefore, we do not adopt the LOO arm selection strategy as the default in COBRA, as the LOO strategy under-performs the standard arm selection strategy currently used in COBRA when all agents report truthfully. LOOM does not work in cases involving complex strategic behavior by agents, such as when multiple agents collude or misreport together. Addressing these challenges by designing new mechanisms will be a promising area of research at the intersection of mechanism design and contextual bandits. We will leave studying these complex settings to future work, for which our work can serve as a foundation.

# E   ADDITIONAL EXPERIMENTS AND DETAILS

This section provides additional details from Section 5, followed by further experimental results.

**Strategic manipulations via feature adaptation.**   We want to highlight that our proposed algorithm, COBRA, operates without prior knowledge of the specific nature of these manipulations, which we model as equivalent to over-reporting. Under the assumption that the agent engages only in strategic over-reporting, the objective is to identify any such over-reporting behavior. For feature adaptation, the agent can strategically manipulate and optimize its features against the deployed algorithm using only binary feedback (whether it was selected or not) in each round as follows: Assume $x_n^\star$ be the true arm-features of agent $n$. The agent can over-report a feature $\tilde{x} = x_n^\star + \eta\Delta$, where $\Delta \in \mathbb{R}^d$ is a bounded perturbation such that $||\Delta||_2 \leq \Delta_{\max}$ and $\eta$ is the learning rate. The agent's goal is to learn a manipulation strategy $\Delta$ using binary feedback that increases its probability of being selected. To do so, the agent uses finite-difference stochastic gradient ascent update.

**Experiments with non-linear reward.**   We also compare the performance of our proposed algorithm, COBRA, for contextual bandit problems with non-linear reward functions.  For this experiment, we adapt problem instances with non-linear reward functions from those used for linear functions in Section 5.  We apply a polynomial kernel of degree 2 to transform the item-agent feature vectors to introduce non-linearity.  The constant terms (i.e., the 1's) resulting from this transformation are removed.  As an example, a sample 4-d feature vector $x = (x_1, x_2, x_3, x_4)$ is transformed into a 14-d feature vector: $x' = (x_1, x_2, x_3, x_4, x_1x_2, x_1x_3, x_1x_4, x_2x_3, x_2x_4, x_3x_4, x_1x_2x_3, x_1x_2x_4, x_1x_3x_4, x_2x_3x_4)$.  We also

remove 1's, which appear in the transformed samples. As expected, our algorithms COBRA based on UCB and TS-based contextual nonlinear bandit algorithms (prefixed with 'n') outperform all the baselines (adapted to non-linear setting, also prefixed with 'n') as shown in Fig. 4. These results are observed across various problem instances, where only the reward function is varied while all other parameters remain unchanged, except for the number of rounds, which is set to $T = 2000$. We further observe that COBRA with TS outperforms its UCB-based counterpart.

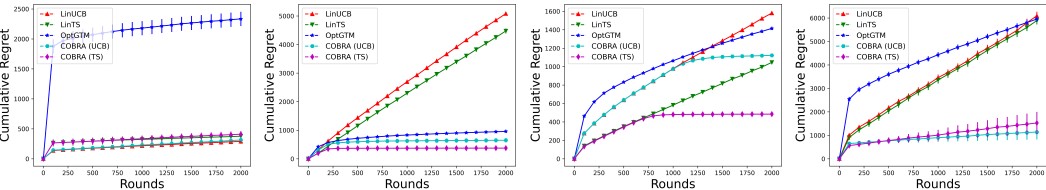

(a) Linear (Truthful setting)  (b) Linear (Agent type I)  (c) Linear (Agent type II)  (d) Square (Agent type II)

Figure 4: Comparing the cumulative regret of COBRA with different baselines for problem instances with non-linear reward functions.

**Regret of COBRA vs. number of agents ($N$) and dimension ($d$).** The number of agents ($N$) and dimension of context-agent feature vector ($2d$) in the contextual bandit problem control the difficulty. As their values increase, the problem becomes more difficult, making it harder to allocate the context to the best agent. We want to verify this by observing how the regret of our proposed algorithms changes while varying $N$ and $d$ in the contextual bandit problem. To see this in our experiments, we use the linear reward function (i.e., $f(x) = 5x^\top \theta_\star$), 2000 contexts, $N = 10$ when varying dimension, $d = 20$ while varying the number of agents. As shown in Fig. 5a and Fig. 5b, the regret bound of our COBRA UCB- and TS- based algorithms increases as we increase the number of agents, i.e., $N = \{10, 20, 30, 40, 50\}$. We also observe the same trend when we increase the dimension of the context-agent feature vector from $d = \{5, 10, 15, 20, 25\}$ as shown in Fig. 5c and Fig. 5d. In all experiments, we also observe that the COBRA TS-based algorithm performs better than its COBRA UCB-based counterpart (as seen in Fig. 5a-5d by comparing the regret of both algorithms).

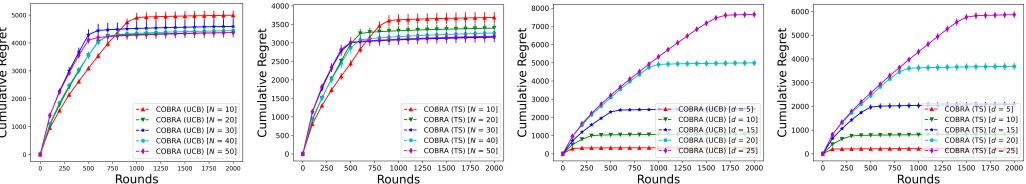

(a) Vary agents/arms (UCB) (b) Vary agents/arms (TS) (c) Vary dimension (UCB)  (d) Vary dimension (TS)

Figure 5: Cumulative regret of COBRA vs. different values of $N$ and $d$.

**Experiments with two strategic agents.** We also conducted additional experiments involving two type II strategic agents. In these experiments, we control the degree of over-reporting for one of the two strategic agents as $\hat{\Delta}_x \sim \mathbb{N}(\text{Scale} * \Delta_x^\star, \sigma_\Delta^2)$. The results (Fig. 6a–6d) show that as the controlled over-reporting of the agent increases, the regret of COBRA also increases. The higher regret is because the strategic behavior becomes harder to detect, and the over-reporting biases the estimator used in arm selection. For the same settings, we also plot the maximum value of $C$ under different levels of over-reporting for the two strategic agents. As expected, $C$ grows very slowly with the number of rounds and remains almost constant, as shown in Fig. 6e–6h.

**Regret of COBRA vs. large number of agents ($N$) and dimension ($d$).** We conducted additional experiments to evaluate the performance of our LOOM-based mechanism in higher-dimensional settings and with larger numbers of strategic agents. Specifically, we considered the following scales: number of agents $N \in 100, 200, 300, 400, 500$ and feature dimension $d \in 60, 70, 80, 90, 100$. Note that overall dimension of context-arm feature is $2d$. When varying the number of agents, we fixed the feature dimension to $50$ and $100$. As expected, the regret of COBRA remains sub-linear as the number of agents increases (Fig. 7a-7b). Similarly, when varying the feature dimension, we set the number of agents to $50$ and $100$. As anticipated, the regret of COBRA increases with the dimension, since higher-dimensional problems require substantially more samples to estimate the reward function accurately (Fig. 7c-7d). These results demonstrate the robustness of our theoretical guarantees, even

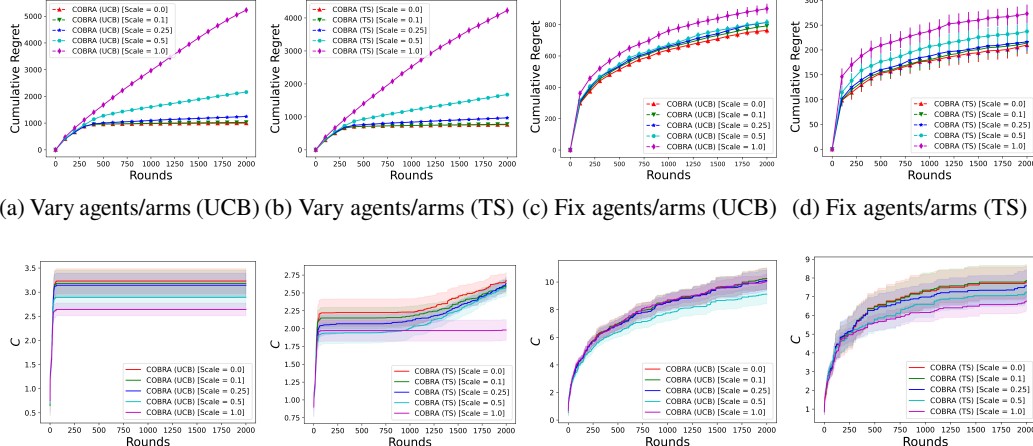

(a) Vary agents/arms (UCB) (b) Vary agents/arms (TS) (c) Fix agents/arms (UCB)  (d) Fix agents/arms (TS)

(e) Vary agents/arms (UCB) (f) Vary agents/arms (TS) (g) Fix agents/arms (UCB)  (h) Fix agents/arms (TS)

Figure 6: **Top row:** Cumulative regret of COBRA vs. different level of two over-reporting agents. **Bottom row:** Maximum value of $C$ vs. different level of two over-reporting agents.

in complex, large-scale environments. For the same settings, we also plot the maximum value of $C$ under different $N$ and $d$. As expected, $C$ grows very slowly with the number of rounds and remains almost constant, as shown in Fig. 7e–7h.

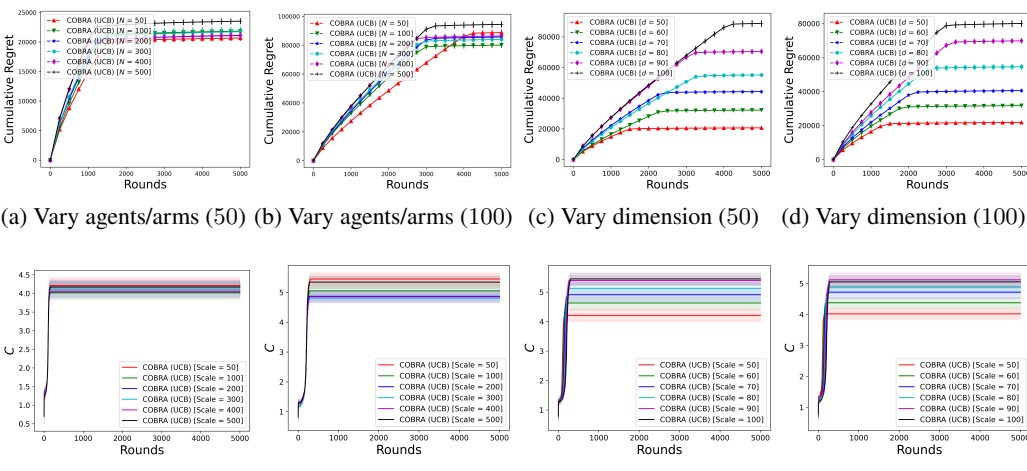

(a) Vary agents/arms (50) (b) Vary agents/arms (100) (c) Vary dimension (50) (d) Vary dimension (100)

(e) Vary agents/arms (50) (f) Vary agents/arms (100) (g) Vary dimension (50) (h) Vary dimension (100)

Figure 7: **Top row:** Cumulative regret of COBRA vs. different values of $N$ and $d$. **Bottom row:** Maximum value of $C$ vs. different values of $N$ and $d$. $(A)$ in captions implies either $N = A$ or $d = A$.

**Computational resources.** All the experiments are run on a Apple M3 Pro with 18GB memory.

**Time and space complexity of COBRA.** The computational complexity of COBRA is comparable to that of standard contextual bandit algorithms, and it scales efficiently even when agents control multiple arms, as discussed below:

**Time complexity:** The overall time complexity is dominated by the underlying estimation procedure, which is identical to that of standard contextual bandit algorithms. LOOM requires computing a Leave-One-Out (LOO) estimator for each agent. These LOO estimators are inherently parallelizable, i.e., one can independently estimate each agent-specific estimator (trained on data excluding agent $i$). This parallelism substantially alleviates computational burden, although the total computational resources required still scale linearly with the number of agents.

**Space complexity:** The space complexity is likewise similar to standard contextual bandit approaches, with the main additional overhead arising from maintaining $N$ separate LOO estimators.

