# OpenReview forum: "COBRA: Contextual Bandit Algorithm for Ensuring Truthful Strategic Agents"
_ICLR.cc/2026/Conference — Submitted to ICLR 2026_

### Official Review · Reviewer_hcLK · 2025-10-28

**Soundness:** 1
**Presentation:** 2
**Contribution:** 1
**Rating:** 0
**Confidence:** 3

**Summary:**

**Disclaimer**: I reviewed this same paper for NeurIPS 2025. The main reason that this paper gets rejected from NeurIPS 2025 is because there seem some fundamental flaws in its approaches and proofs and some explicit proof bugs were also identified (see my detailed description below). There was a very long technical discussion during the rebuttal phase of NeurIPS 2025, but ultimately the discussion did not resolve the technical issues/bugs identified by the reviewers there. The ICLR 2026 draft did a bit re-writing, e.g., by re-phrasing some assumptions into a “LOOM-compatible” definition, but the rephrasing did not seem to change the earlier issues and just make it appear differently.

This paper develops a new algorithm for strategic contextual bandit problem where each arm is a strategic agent and can misreport their feature vector, though cannot manipulate reward. This problem was proposed and studied by Kleine Buening et al. NeurIPS'24, but the current paper develops a different algorithm inspired by VCG. Specifically, the paper proposes to use all agents' information, excluding arm a's, to estimate a reward function for agent a (in linear bandit case, this means estimate the theta parameter for a). The paper argues that this, coupled with an optimistic-pessimistic inequality called LOOM condition, can help to induce incentive compatibility.

**Strengths:**

Interesting research question.

**Weaknesses:**

The paper’s writing has some clarity issues. For instance, in the definition of “LOOM-compatible contextual bandit algorithm”, it is unclear why “any contextual bandit algorithm” would always have an “estimated function”. We know UCB has a natural estimate function of its reward, but if we do Thomas Sampling, do we call that updated distribution mean as the “estimate”? What’s the definition of “estimated function”? How do you ensure that “any” bandit algorithm will have an “estimated function”?

What is the difference between Theorem 1 and Theorem 4?

The above are some smaller technical issues I found, but **my biggest concern is that there seem major flaws in the proposed approach, as well as in the technical proofs,** which makes the paper not acceptable. In particular, the paper's approach of using all other agent's information to estimate a parameter and reward for arm a could not work for linear contextual bandit. The no regret proof for standard linear context bandit, which this paper adapts from, crucially depends on a "self-normalization lemma", which roughly says if we use data along some direction d to estimate parameter theta, then in the future when we see feature roughly along direction d, our reward estimation error will be small. However, this paper's approach of using other arms' information to estimate i's arm reward basically excluded the possibility of such "self-normalization".

**A potential counterexample**  Let us consider the standard multi-armed bandit as a special case of linear contextual bandit. This is a special case because we can view x_{a,t} as always along the canonical basis e_a direction, with rescaling factor as agent’s private information at each round t. In such special cases, LinUCB algorithm effectively becomes estimating the parameter theta_a (a'th dimension of theta) along direction e_a separately and independently using arm a's previous information. This can be verified by plugging in x_{a,t} as a scaled version of e_a into LinUCB descriptions in Line 300 to 306. However, using this paper's proposed approach, as explained in LOOM Condition in Equation (2) and algorithm description of COBRA, COBRA will use information orthogonal to direction e_a to decide whether e_a’s report is correct or not. This simply is impossible since the information from any arm other than a will not be useful for estimating \theta_a. Concretely in Equation (2), the LCB for an arm a, estimated used all other arms orthogonal to a, will always much larger than the UCB on the right hand side, and will never change over the execution of the algorithm.

The paper’s technical writing is a bit difficult to follow, but I tried to check the detailed proof to see where concrete proof bugs may appear in the paper. I think one concrete bug is the following, inherently due to the issue I mentioned above.

 In the proof of Theorem 4 in the Appendix, I think the inequality from Line 1102 to Line 1105 is incorrect. The paper claims that a Det/Det term is always upper bounded by a constant C. This is not correct. In the standard multi-armed bandit special case where arms are canonic basis vectors, this term is lower bounded by the number of times arm a is pulled, which can be \Omega(T) and cannot be upper bounded by a universal constant C. That makes the regret linear.

Another thing that strikes me as odd when diving deep into the proof of Theorem 4 is that I don’t see where they use the LOOM strategy of banning a misreported arm at all. If the proof were correct, they could arrive at the regret bound in equation (18) without ever using an argument about the NE and the regret bound would hold for ANY arm strategy?! I might be missing something here, but otherwise this obviously can’t be true (or their assumptions are overly strong so that there is generally no point in doing mechanism design in their setting).

**Questions:**

Feel free to respond to my concerns about proof bugs above. I am happy to be convinced and revise my ratings, but currently I do not see a way to overcome the intrinsic issue of the approach.

**Details Of Ethics Concerns:**

I am a bit concerned that the paper was re-submitted without any major changes to the originally flawed proofs.

---

> ### Author Response · Authors · 2025-12-04
> **Part I**
>
> We would like to thank the reviewer for taking the time to carefully review our paper. We have addressed the weakness that you raised and answered your question as follows:
>
>
> > **The paper’s writing has some clarity issues. For instance, in the definition of “LOOM-compatible contextual bandit algorithm”, it is unclear why “any contextual bandit algorithm” would always have an “estimated function”. We know UCB has a natural estimate function of its reward, but if we do Thomas Sampling, do we call that updated distribution mean as the “estimate”? What’s the definition of “estimated function”? How do you ensure that “any” bandit algorithm will have an “estimated function”?**
>
> Many contextual bandit algorithms bound the instantaneous regret, which can be expressed in the form $ $h(x_{t,a}, O_{t-1}) = D\alpha_T ||x_{t,a}||_{V_{t-1}^{-1}} + E$ $, where $D$ and $E$ are constants that do not depend on the round. For example, for Lin-UCB, $ $h(x_{t,a}, O_{t-1}) = \alpha_T ||x_{t,a}||_{V_{t-1}^{-1}} + \sqrt{\lambda} S$ $, as shown in Table 2 in the Appendix (Page 24). As long as the sum $\sum_{t=1}^T h(x_{t,a}, O_{t-1})$ is upper bounded by a sub-linear function of $T$ for any contextual bandit algorithm, whether it uses UCB or Thompson sampling (TS) for arm selection, COBRA also achieves sub-linear regret. We refer to such algorithms as LOOM-compatible algorithms.
>
> The "estimated function" represents the estimate of the underlying reward function. Both UCB- and TS-based contextual bandit algorithms use this estimate for arm selection. However, if a contextual bandit algorithm does not belong to the class of LOOM-compatible algorithms, our regret bounds (Thoerems 3 and 5) may not directly apply when that algorithm is plugged into COBRA.
>
>
> > **What is the difference between Theorem 1 and Theorem 4?**
>
> Theorem 1 shows that LOOM does not incorrectly identify an agent as misreporting with high probability, whereas Theorem 4 quantifies the regret of COBRA (using Lin-UCB) when any agent over-reports.
>
> > **The above are some smaller technical issues I found, but **my biggest concern is that there seem major flaws in the proposed approach, as well as in the technical proofs**, which makes the paper not acceptable. In particular, the paper's approach of using all other agent's information to estimate a parameter and reward for arm a could not work for linear contextual bandit. The no regret proof for standard linear context bandit, which this paper adapts from, crucially depends on a "self-normalization lemma", which roughly says if we use data along some direction d to estimate parameter theta, then in the future when we see feature roughly along direction d, our reward estimation error will be small. However, this paper's approach of using other arms' information to estimate i's arm reward basically excluded the possibility of such "self-normalization".**
>
> We agree with the reviewer that self-normalization is not possible when agents are heterogeneous. However, COBRA consists of two components: arm selection strategy and LOOM. We discuss how heterogeneous agents affect these components:
>
> - **Arm selection strategy:** COBRA uses all available information to estimate the reward function and calculate exploration bonuses for arm selection, which is equivalent to a standard contextual bandit setting. The presence of heterogeneous agents does not directly affect the self-normalization bound. However, misreported arm features can bias the reward function estimator and degrade exploration bonuses, potentially leading to the selection of suboptimal agents, typically those that overreport. It is difficult to derive theoretical guarantees in this setting without additional assumptions. Therefore, we make Assumption 1, which is similar to prior work, as discussed in Appendix D.5 (Page 27). Assumption 1 ensures that agents are neither too heterogeneous nor too homogeneous, allowing the analysis to hold.
> - **LOOM component:** To analyze LOOM, we need to assume that the constant $C$ is bounded. $C$ captures the complexity of the problem, specifically the heterogeneity of agents.
>
> As long as Assumption 1 holds, we can show that COBRA achieves sub-linear regret.

---

> ### Author Response · Authors · 2025-12-04
> **Part II**
>
> > **A potential counter example Let us consider the standard multi-armed bandit as a special case of linear contextual bandit. This is a special case because we can view $x_{a,t}$ as always along the canonical basis $e_a$ direction, with rescaling factor as agent’s private information at each round $t$. In such special cases, LinUCB algorithm effectively becomes estimating the parameter theta_a (a'th dimension of theta) along direction e_a separately and independently using arm $a$'s previous information. This can be verified by plugging in $x_{a,t}$ as a scaled version of $e_a$ into LinUCB descriptions in Line 300 to 306. However, using this paper's proposed approach, as explained in LOOM Condition in Equation (2) and algorithm description of COBRA, COBRA will use information orthogonal to direction $e_a$ to decide whether $e_a$’s report is correct or not. This simply is impossible since the information from any arm other than a will not be useful for estimating $\theta_a$. Concretely in Equation (2), the LCB for an arm $a$, estimated used all other arms orthogonal to $a$, will always much larger than the UCB on the right hand side, and will never change over the execution of the algorithm.**
>
> We respectfully disagree with the reviewer, as this example does not necessarily lead to the failure of COBRA. In standard multi-armed bandits, the $a$-th dimension of $\theta$ represents the estimated mean reward of arm $a$. In our setting, learners observe noisy samples of the true reward, so the mean reward estimation of each agent/arm is independent of that of other agents/arms. Consequently, over-reporting by scaling the canonical basis in the $e_a$ direction does not affect the mean reward estimation of an arm, since the learner only considers how many times each agent is selected. Furthermore, COBRA continues to operate correctly even when the learner uses the reported context of a strategic agent, because we assume that context–arm features are bounded (Line 148), a standard assumption in contextual bandits. The bounded feature vectors limit the magnitude of any feature scaling by the strategic agent, and the estimated reward function parameters are scaled accordingly without affecting the underlying mean reward estimation. Therefore, the arm selection strategy cannot be manipulated effectively by over-reporting.
>
> > **The paper’s technical writing is a bit difficult to follow, but I tried to check the detailed proof to see where concrete proof bugs may appear in the paper. I think one concrete bug is the following, inherently due to the issue I mentioned above. In the proof of Theorem 4 in the Appendix, I think the inequality from Line 1102 to Line 1105 is incorrect. The paper claims that a Det/Det term is always upper bounded by a constant C. This is not correct. In the standard multi-armed bandit special case where arms are canonic basis vectors, this term is lower bounded by the number of times arm a is pulled, which can be $\Omega(T)$ and cannot be upper bounded by a universal constant $C$. That makes the regret linear.**
>
> Our goal was to obtain an upper bound for $ $\frac{||x_{t,a}||_{V^{-1}_{t,-a}}}{||x_{t,a}||_{V^{-1}_{t}}}$ $. We agree with the reviewer that this bound in the current approach may become so loose that it could make the overall regret linear. Therefore, we have updated the proof to directly use $C$ as an upper bound for $ $\frac{||x_{t,a}||_{V^{-1}_{t,-a}}}{||x_{t,a}||_{V^{-1}_{t}}}$ $.
> Although we are not able to provide a theoretical bound for $C$, we have empirically shown that, in both homogeneous settings (where arm features are selected randomly for each context) and heterogeneous settings (where arm features are fixed for all contexts), $C$ grows only slowly (almost constant) with the number of rounds in our considered settings.
>
>
> > **Another thing that strikes me as odd when diving deep into the proof of Theorem 4 is that I don’t see where they use the LOOM strategy of banning a misreported arm at all. If the proof were correct, they could arrive at the regret bound in equation (18) without ever using an argument about the NE and the regret bound would hold for ANY arm strategy?! I might be missing something here, but otherwise this obviously can’t be true (or their assumptions are overly strong so that there is generally no point in doing mechanism design in their setting).**
>
> We would like to highlight that LOOM is explicitly used in the proof of our regret bound on page 21. In particular, *Lemma 4 leverages the LOOM condition* to derive an upper bound on the maximum gain an agent can achieve by misreporting arm features before being detected by LOOM.

---

### Official Review · Reviewer_VEb8 · 2025-10-31

**Soundness:** 1
**Presentation:** 2
**Contribution:** 2
**Rating:** 2
**Confidence:** 4

**Summary:**

The paper considers a contextual bandit problem with strategic arns who misreport contexts to maximize their number of selections over $T$ rounds. The authors propose a VCG-inspired algorithm that uses optimistic and pessimistic estimates of each agent’s reward to eliminate misreporting agents. It is shown that under the proposed algorithm truthfulness is an approximate Nash equilibrium and the authors establish sublinear regret bounds given that the arms play in NE. The authors also provide empirical results on synthetic problem instances to support their theoretical claims.

**Strengths:**

1. The studied problem is interesting, and the intersection of online regret minimization under uncertainty with mechanism design is a challenging but interesting domain.
2. The authors motivate the model and the work well.

**Weaknesses:**

I have concerns about the correctness of Theorem 4. Firstly, there a various typos in Appendix B.2.2 which make the proof of Theorem 4 hard to read. For example, Lemma 4 has various typos and it is unclear  what $a_a$ is, what the $x$ in the definition of $UCB_{t, -a} (x_{s, a_a})$ in line 996 is, etc.

Following this, I am confused about line 1067 in the proof. You plug-in Lemma 4, but what is the $x$ in $\lVert x \rVert_{V_{t, -a}^{-1}}$. As far as I can tell, this $x$ should be $x_{t,a}$. However, then bounding $\sum_{t=1}^T \lVert x_{t,a} \rVert_{V_{t,-a}^{-1}}$ is hard and I believe that you shouldn't be able to bound this the way you do. Consider the case where the context vectors of arm $a$ are linearly independent of the context vectors of all other arms $a'$ (for all time steps). Then, there is no good bound for the sum over $\lVert x_{t,a} \rVert_{V_{t,-a}^{-1}}$. In other words, if you are only using every other arm's data, your exploration bonueses can stay arbitrarily large. I have difficulties following the reasoning on the top of page 21 (partly due to typos), but I suspect that the issue is there.

Another sign that something is possibly wrong is that you arrive at your regret bound (18) on page 21 without ever using LOOM or any guarantee about the arm strategies. What would be the point of mechanism design when you can get the same regret guarantee without using that the arms play a NE under COBRA?

Other weaknesses:
- The presentation could be improved, and particularly Section 4 is difficult to read as it is extremely dense. Related to the issues in the presentation, you introduce COBRA(TS) and then present your theoretical guarantees as if they also hold for COBRA(TS). As far as I can tell you only prove results for COBRA(UCB); see e.g., proof of Theorem 2. It is often quite unclear what algorithm you are referring to when you just write COBRA.

I'd be happy to increase my score if my concerns can be resolved.

**Questions:**

1. In line 131, you say that the previous work's [1] method may not be practical when the true reward function is unknown. Could you please be precise about what you mean by this? The algorithm in [1] appears to be designed specifically for the case where the true reward function is unknown.
2. Line 336: It should be COBRA(TS) instead of COBRA(UCB).

[1] Thomas Kleine Buening, Aadirupa Saha, Christos Dimitrakakis, Haifeng Xu; Strategic Linear Contextual Bandits, NeurIPS 2024

---

> ### Author Response · Authors · 2025-12-04
> **Part I**
>
> We would like to thank the reviewer for taking the time to carefully review our paper. We have addressed the weakness that you raised and answered your question as follows:
>
>
> > **I have concerns about the correctness of Theorem 4. Firstly, there a various typos in Appendix B.2.2 which make the proof of Theorem 4 hard to read. For example, Lemma 4 has various typos and it is unclear what is $a_a$, what the $x$ in the definition of $UCB_{t,-a}(x_s,a_s)$ in line 996 is, etc.**
>
> Thank you for pointing out the typos. We have corrected them in the revised paper. Specifically, $a_a$ has been replaced with $a$, and $x$  has been replaced with $x_{s,a}$, where $a$ denotes the agent selected in round $s$  (i.e., $a_s = a$), and $x_{s,a}$ represents the context–arm feature vector corresponding to agent $a$ in that round $s$.
>
>
> > **Following this, I am confused about line 1067 in the proof. You plug-in Lemma 4, but what is the $x$ in $ $\|\|x\|\|_{V_{t,-a}^{-1}} $ $. As far as I can tell, this $x$ should be $x_{t,a}$.**
>
> The reviewer is correct: $x$ should be $x_{t,a}$. This typo has been corrected in the updated version of the paper.
>
> > **However, then bounding $ $ \sum_{t=1}^{T}\|\|x_{t,a}\|\|_{V^{-1}_{t,-a}} $ $ is hard and I believe that you shouldn't be able to bound this the way you do. Consider the case where the context vectors of arm  $a$ are linearly independent of the context vectors of all other arms $a'$ (for all time steps). Then, there is no good bound for the sum over  $ $\|\|x_{t,a}\|\|_{V^{-1}_{t,-a}}$ $. In other words, if you are only using every other arm's data, your exploration bonueses can stay arbitrarily large. I have difficulties following the reasoning on the top of page 21 (partly due to typos), but I suspect that the issue is there.**
>
> In the following, we address your comments:
>
> - **Bounding $ $\sum_{t=1}^{T}||x_{t,a}||_{V^{-1}_{t,-a}}$ $:** We agree with the reviewer that bounding $ $\sum_{t=1}^{T}||x_{t,a}||_{V^{-1}_{t,-a}}$ $ is difficult. This is why, in the proof (Lines 1092-1093), we multiply and divide each term by $ $||x_{t,a}||_{V^{-1}_{t}}$ $ and then use an upper bound for $ $\frac{||x_{t,a}||_{V^{-1}_{t,-a}}}{||x_{t,a}||_{V^{-1}_{t}}}$ $, which we denote by $C$ in Line 1077. The constant $C$ also reflects how similar an agent is to the others (a smaller value indicates greater similarity). Although we are not able to theoretically bound the value of $C$, we have empirically validated the maximum value of $C$ in our experiments (shown in Figures 6 and 7 in the Appendix E). Furthermore, the value of $C$ only influences the analysis when COBRA selects a suboptimal arm. Moreover, COBRA uses all context–arm feature vectors and corresponding rewards from all agents when selecting an arm, making this step equivalent to a standard contextual bandit algorithm. As discussed in Appendix D.1, when there is only a single unique agent, there is no incentive for that agent to misreport.
>
> - **Arbitrarily large exploration bonueses in LOOM:** We agree with the reviewer that the exploration bonuses $UCB_{t,-a}(x_{s,a})$ can become large. When agents are not sufficiently homogeneous, the other agents may fail to detect whether a given agent is misreporting. However, this does not affect the standard arm-selection component of the algorithm; it only impacts the LOOM procedure. In particular, LOOM may fail to satisfy Eq. (2) (Line 269), and consequently, COBRA may be unable to identify the agent that is strategically misreporting. We have conducted additional experiments in which agents are completely heterogeneous, using fixed arm-feature vectors for each agent (see Figure 6 in Appendix E). These experiments demonstrate that COBRA continues to achieve sub-linear regret, as the observed rewards are not affected by misreported arm features. However, as expected, LOOM is unable to identify the strategic agent in this heterogeneous setting.
>
>
> > **Another sign that something is possibly wrong is that you arrive at your regret bound (18) on page 21 without ever using LOOM or any guarantee about the arm strategies. What would be the point of mechanism design when you can get the same regret guarantee without using that the arms play a NE under COBRA?**
>
> We would like to highlight that LOOM is explicitly used in the regret analysis on Page 20. In particular, our proof relies on Lemma 4 (Lines 1066-1067), which leverages the LOOM condition to derive an upper bound on the maximum gain an agent can obtain by misreporting arm features before being detected by LOOM.

---

> > ### Author Response · Authors · 2025-12-04
> > **Part II**
> >
> > > **The presentation could be improved, and particularly Section 4 is difficult to read as it is extremely dense. Related to the issues in the presentation, you introduce COBRA(TS) and then present your theoretical guarantees as if they also hold for COBRA(TS). As far as I can tell you only prove results for COBRA(UCB); see e.g., proof of Theorem 2. It is often quite unclear what algorithm you are referring to when you just write COBRA.**
> >
> > Our COBRA algorithm can take any LOOM-compatible contextual bandit algorithm as input. Depending on the problem setting, COBRA allows the use of different contextual bandit algorithms, such as Lin-UCB, LinTS, Neural-UCB, Neural-TS, and others. To make this clearer, we use the notation COBRA($A$), where $A$ denotes the underlying contextual bandit algorithm.
> >
> > Since the goal of Section 4 was to convey the high-level ideas, we chose a linear reward function and UCB as the selection strategy. For brevity, we ignore this notation when introducing COBRA with Lin-UCB as a contextual bandit algorithm.
> >
> >
> > > **In line 131, you say that the previous work's [1] method may not be practical when the true reward function is unknown. Could you please be precise about what you mean by this? The algorithm in [1] appears to be designed specifically for the case where the true reward function is unknown.**
> >
> > The previous work [1] proposed two algorithms. In the first algorithm, it is assumed that the reward function parameters are known. Under this assumption, one can use only the agent’s reported context–arm features, together with the known reward function, to compare with noisy rewards. However, this approach does not extend to cases where the underlying reward function parameters are unknown, as there is no baseline to determine whether an agent is misreporting using only their own reported arm features. Consequently, the method in [1] may fail to identify a strategic agent.
> > In contrast, our LOOM mechanism, inspired by VCG, leverages reports from other agents to detect misreporting without requiring agent-specific baselines. While both works consider linear rewards and strategic agents, our formulation explicitly incorporates incentive compatibility through LOOM and can support non-linear extensions, yielding stronger and more practical guarantees.
> > As discussed in Appendix D.1, we note that LOOM still has limitations in identifying strategic agents in certain settings; addressing these limitations is left as future work.
> >
> > > **Line 336: It should be COBRA(TS) instead of COBRA(UCB).**
> >
> > Thank you for catching the typo. It has been corrected to COBRA (TS) in the revised paper.
> >
> > 1. Kleine Buening, Thomas, Aadirupa Saha, Christos Dimitrakakis, and Haifeng Xu. "Strategic linear contextual bandits." Advances in Neural Information Processing Systems 37 (2024): 116638-116675.

---

### Official Review · Reviewer_BSfu · 2025-11-02

**Soundness:** 3
**Presentation:** 2
**Contribution:** 3
**Rating:** 6
**Confidence:** 4

**Summary:**

This paper considers incentive-copmatible contextual bandits, where one learner is interacting with strategic agents controlling the arms' contexts.

After observing the context $c_t$, the learner selects $a_t$, resulting in a stochastic reward with mean $f(x_{t,a_t})$ where $x_{t,a} = \phi(c_t, a) \in R^d$ is the feature vector associated with $c_t$.
It is unclear, but I assume that neither $c_t$ or $\phi$ are known to the agent.

A linear version of this problem was previously studied by Buening et al, but the techniques used in this paper are substantially different.

The main idea is to test if agents are over-reporting and exlude them. The threat of exclusion provides the incentive. For that reason they define the following estimates:  $f_t$, which is based on everybody's estimate, and $f_{t,-a}$ which excludes $a$'s reports.

The algorithm *could* be combined with a large class of context bandit algorithms. Of course, this requires carefully looking whether various conditions are satisfied. This is captured in Assumption 1 where they asume that:
  (a) $f(x) \leq UCB_{t,a}(x)$,
  (b) $UCB_t(x_{t,a}) \leq UCB_{t,-a}(x_{t,a})$
where
$UCB_{t,a}(x) = f_{t,a}(x) + \epsilon_t$
$UCB_{t,-a}(x) =  f_{t,-a}(x) + \epsilon_{t,-a}$,
with $\epsilon$ denoting appropriate confidence intervals around estimates.

The assumption is examined page 30. Case 2, where one agent over-reports, is the basic scenario. But given, how central this assumption is to the proofs, this is really an inadequate proof for me. I guess the $-a$ interval should be wider than the $a$ one, but this should be proven more rigorously.

The other quantity is $LCB$, which structured differently:
- $LCB_{t,-a} = f_{t,-a} - \epsilon_{t,-a}$.
- $LCB^{(x)}_{t,a} = \sum_{s=1, a_s = a}^t LCB_{t,-a}(x_{s,a_s})$,
i.e. it is the sum of lower bounds through other agents reports, where $x$ now defines a sequence of rewards (confusingly).

This is complemented with $UCB^{(y)}$, an upper bound on the total reward which does *not* use the contexts. Perhaps this notation is a bit counterintuitive. In any case the  LOOM condition can be summarised as follows:

If the lower bound, constructed through context-dependent estimates using agents reports, exceeds the upper bound constructed only through observed rewards from agent $a$, then $a$ is over-reporting.

**Strengths:**

+ The main idea is nice and intuitive
+ The results are an improvement and extension of previous work

**Weaknesses:**

- The presentation could be improved. I spent more time to understand what is going than I should have had to.
- Assumption 1 can be quite restrictive. It should at least be cleanly proven for some special cases, but the discussion in Appendix D is inadequate. Intuitively, it should hold for the linear case based on the reasoning given.

**Questions:**

? The paper also says it is inspired by VCG, but the connection is not
spelled out. I assume it is because $f_{t,-a}$ uses the reports of the
remaining agents.

---

> ### Author Response · Authors · 2025-12-04
>
> We would like to thank the reviewer for taking the time to carefully review our paper. We have addressed the weakness that you raised and answered your question as follows:
>
>
> > **Assumption 1 can be quite restrictive. It should at least be cleanly proven for some special cases, but the discussion in Appendix D is inadequate. Intuitively, it should hold for the linear case based on the reasoning given.**
>
> We would like to emphasize that regret guarantees are established for the case in which all agents report truthfully (Theorems 2 (linear reward function) and 3 (non-linear reward function)). In contrast, when agents are allowed to over-report, deriving theoretical guarantees becomes challenging without additional assumptions. Assumption 1 allows us to bound the regret while using LOOM for the identification of strategic agents who over-report. We have also shown that our assumption is no stronger than those commonly used in existing work.
>
>
> > **The paper also says it is inspired by VCG, but the connection is not spelled out. I assume it is because $f_{t,-a}$ uses the reports of the remaining agents.**
>
> Our LOOM mechanism, inspired by VCG, uses context-arm features reported by other agents to detect over-reporting agents. While prior work also considers linear rewards and strategic agents, our formulation explicitly incorporates incentive compatibility through LOOM and can support non-linear extensions, yielding stronger and more practical guarantees. When verifying incentive compatibility, LOOM compares optimistic and pessimistic estimates based on the reports of other agents (i.e., excluding agent $a$ when evaluating $f_{t,-a}$. This approach is conceptually similar to VCG-type mechanisms, although no payments are involved in our setting.

---

### Official Review · Reviewer_acmw · 2025-11-05

**Soundness:** 3
**Presentation:** 2
**Contribution:** 2
**Rating:** 6
**Confidence:** 3

**Summary:**

This paper addresses contextual bandit problems where strategic agents may misreport their features to maximize their selection probability. The authors propose COBRA (Contextual Bandit Algorithm for Ensuring Truthful Strategic Agents), which uses a Leave-One-Out-based Mechanism (LOOM) inspired by VCG mechanisms to detect and disincentivize misreporting. The authors propose that reporting arm features truthfully is the best/dominant strategy for the agents which is achieved via LOOM and COBRA. Experimental results validate their theoretical results of sublinear regret.

**Strengths:**

1) Misreporting in contextual bandits is clearly motivated (food delivery/marketplace settings) and is an interesting/practically relevant problem.
2) LOOM provides a theoretically grounded, drop-in mechanism compatible with common contextual bandit algorithms.
3) The proofs in the appendix are well structured.

**Weaknesses:**

1) Only synthetic evaluation. Considering that real world applications are well motivated and reiterated throughout the paper it would have been nice to see some experiments on real world data.
2) The scale of the synthetic experiments is quite small as well. Having just 5 agents with only one of the agents over reporting (line 465) is a bit unsatisfactory in terms of scale. It would be better to see experiments on a larger scale particularly larger $d$ and $N$ than those found in the appendix, and with more than one over reporter.
3) A limitations sections would benefit the paper. The term dominant strategy is only applicable for the case of over reporting and can be misleading considering the paper assumes that collusion and under reporting cannot happen (which are strong assumptions in their own right). The failure conditions of LOOM should also be mentioned in the main paper instead of being scattered around in remarks and in the appendix.
4) The complexity of LOOM + COBRA should be discussed, especially in the case of agents with multiple arms. It would be great if the authors could go into more detail about how their algorithm scales to agents with multiple arms instead of the short paragraph in the appendix to determine application feasibility.

**Questions:**

Questions:
1) Modern applications including the ones motivated in the paper routinely handle thousand to millions of items at once. What is the computational complexity of LOOM+COBRA? Considering this and the dependence of the regret on $\sqrt{N}$, do you believe it is applicable to this scenario ?
2)  How does the complexity change if each agent picks multiple arms?
3) Would it possible to add a semi synthetic or small real world experiments at a larger scale $(d \ge 50, N\ge 50$, more than one over reporter) than those in the appendix
4) See 3) from Weaknesses.

Minor Corrections/Presentation issues:
The paper has few typos and grammatical errors
- line 103: non-leaner --> non-linear
- line 113,118: study an --> studied an
- line 239: "Note that the assumptions underlying contextual bandit algorithms need to satisfy in our setting" (should be rephrased)
- line 336: "propose a TS-based variant COBRA(UCB)" should be COBRA(TS)
- line 373: drive --> derive and quite a few more in the appendix.

Additionally it would be great if the authors could state/explain the notation used in Table 1 of section C of the appendix, in the paper itself. I would also prefer if the core equations were set on their own display lines rather than wrapping across text lines.

---

> ### Author Response · Authors · 2025-12-04
> **Part I**
>
> We would like to thank the reviewer for taking the time to carefully review our paper. We have addressed the weakness that you raised and answered your question as follows:
>
>
> > **Only synthetic evaluation. Considering that real world applications are well motivated and reiterated throughout the paper it would have been nice to see some experiments on real world data.
> > Would it possible to add a semi synthetic or small real world experiments**
>
> The use of synthetic problem instances is a standard and common practice in the bandit literature. This approach allows a rigorous and isolated validation of the algorithm's theoretical properties, such as incentive compatibility and regret bounds in our case, which is challenging to achieve with existing real-world datasets that do not align with our specific strategic model. Moreover, there is no publicly available real-world dataset that currently captures the strategic misreporting necessary to accurately evaluate our incentive mechanism (LOOM) in a contextual bandit setting. Evaluating whether our algorithm (COBRA) successfully incentivizes truthful reporting requires the ability to simulate strategic agent behavior, which in turn necessitates a synthetic environment.
>
>
> > **The scale of the synthetic experiments is quite small as well. Having just 5 agents with only one of the agents over reporting (line 465) is a bit unsatisfactory in terms of scale. It would be better to see experiments on a larger scale particularly larger $d$ and $N$ than those found in the appendix, and with more than one over reporter.
> > Would it possible to add a semi synthetic or small real world experiments at a larger scale ($d\ge 50$, $N\ge 50$, more than one over reporter) than those in the appendix.**
>
> We have conducted additional experiments specifically to demonstrate the performance of our LOOM-based mechanism in higher-dimensional settings and larger numbers of strategic agents. We have included new results based on the following larger scales:
> - Dimension ($d$): We now include results for $ d \in \{60,70,80,90,100 \}$ for each context and arm features (overall dimension of context-arm feature is $2d$), while we set the number of agents to $50$ and $100$. As expected, the regret of COBRA increases with the dimension, as higher-dimensional problems require substantially more samples to obtain a sufficiently accurate reward-function estimator.
> - Number of Agents ($N$): We now include results for $N \in \{100,200,300,400,500\}$ when the dimension is set to $50$ and $100$. As expected, the regret of COBRA remains sub-linear as the number of agents increases.
> - Experiments with two strategic agents: We also added additional experiments involving two type II strategic agents. In these experiments, we control the degree of over-reporting for one of two strategic agents. These results show that as the agent's controlled over-reporting increases, the regret of COBRA also increases due to the strategic behavior becoming harder to detect, and the over-reporting makes the estimator used in arm selection biased.
>
> These results (Figures 6 and 7 in Appendix E of the revised paper) demonstrate the robustness of our theoretical guarantees even in complex, large-scale environments.
>
>
> > **A limitations sections would benefit the paper. The term dominant strategy is only applicable for the case of over reporting and can be misleading considering the paper assumes that collusion and under reporting cannot happen (which are strong assumptions in their own right). The failure conditions of LOOM should also be mentioned in the main paper instead of being scattered around in remarks and in the appendix.**
>
> We thank the reviewer for these suggestions. We have briefly mentioned the limitation of LOOM in the main part of the revised paper (Lines 286-287) and added a reference to the appendix section for a detailed discussion.

---

> > ### Author Response · Authors · 2025-12-04
> > **Part II**
> >
> > > **The complexity of LOOM + COBRA should be discussed, especially in the case of agents with multiple arms. It would be great if the authors could go into more detail about how their algorithm scales to agents with multiple arms instead of the short paragraph in the appendix to determine application feasibility.**
> >
> > The computational complexity of COBRA is comparable to that of standard contextual bandit algorithms, and it scales efficiently even when agents control multiple arms, as discussed below:
> > - **Time complexity:** The overall time complexity is dominated by the underlying estimation procedure, which is identical to that of standard contextual bandit algorithms. LOOM requires computing a Leave-One-Out (LOO) estimator for each agent. These LOO estimators are inherently parallelizable, i.e., one can independently estimate each agent-specific estimator (trained on data excluding agent $i$). This parallelism substantially alleviates computational burden, although the total computational resources required still scale linearly with the number of agents.
> > - **Space complexity.** The space complexity is likewise similar to standard contextual bandit approaches, with the main additional overhead arising from maintaining $N$ separate LOO estimators.
> >
> > We have added this discussion to Appendix E of the revised paper.
> >
> >
> > > **Modern applications including the ones motivated in the paper routinely handle thousand to millions of items at once. What is the computational complexity of LOOM+COBRA? Considering this and the dependence of the regret on $\sqrt{N}$, do you believe it is applicable to this scenario?**
> >
> > The applicability of COBRA to scenarios with millions of items depends on how the algorithm is deployed. Although the total catalog may contain millions of items (e.g., all products on an e-commerce platform), the number of relevant arms/agents presented to a user for any specific query (e.g., "polo t-shirt") is typically much smaller. Because the algorithm operates only on this query-dependent subset, its computational complexity scales with the size of the relevant item set rather than the size of the entire catalog. Furthermore, the effective number of relevant agents is typically on the order of hundreds to thousands (e.g., in online platforms), since each agent may be associated with multiple items.
> >
> > We acknowledge that a very large number of agents ($N$) may increase computational complexity and affect regret bounds due to the $\sqrt{N}$ term. Note that we can perform all the agents-related computations in parallel as they are independent of each other.
> >
> >
> > > **How does the complexity change if each agent picks multiple arms?**
> >
> > Our results generalize to the setting where each agent controls and reports multiple arms. The mechanism maintains individual records of each agent's reported feature vectors and reward history for their respective arms, applying LOOM's statistical test independently to each agent. This allows LOOM to detect and reject agents whose optimistic/pessimistic reward gaps exceed the threshold, even if the agent misreports its different arms in distinct ways.
> >
> > Regarding regret and approximate equilibrium guarantees, as long as Assumption 1 holds for each active agent, our results continue to apply. Specifically, the regret bound remains $\tilde{O}(d\sqrt{T}+\sqrt{NT})$, where $N$ denotes the number of active agents (as discussed in Appendix D.1).
> >
> >
> > > **Minor Corrections/Presentation issues:**
> >
> > Thanks for catching typos. We have fixed them in the revised version of the paper.

---

### Meta-Review · Area_Chair_q1N2 · 2026-01-05

**Summary:**

The reviewers think the problem is kind of interesting and motivated, since it’s about contextual bandits and mechanism design, and the idea of discouraging misreporting without payments sounds nice. But there are big doubts about the technical stuff and clarity. Two reviewers especially don’t trust the main regret analysis, especially for linear contextual bandits. They say Theorem 4 and related proofs look wrong in a fundamental way. Issues include missing proper self-normalization when removing an agent’s own data, determinant bounds that seem questionable and might even cause linear regret, and also the regret bound doesn’t clearly depend on the LOOM mechanism or equilibrium behavior. The authors replied with detailed rebuttals and fixed some typos and notation, but the main correctness concerns and strong assumptions are still there. Other reviewers also said assumptions are too restrictive, the writing is dense and confusing sometimes, and experiments are only on synthetic data. So overall, even if the direction is interesting, the technical problems and lack of solid guarantees make it not ready for acceptance.

**Reviewer Concerns:**

Some typos and notation stuff were fixed in the rebuttal, and they added a bit more about limitations and complexity. They also did bigger synthetic experiments, so that partly addresses the scale issue. But the main problems are still there: correctness of the regret analysis, Assumption 1, the linear bandit counterexamples, and whether LOOM is actually used in the proofs in any real way. The explanations don’t really fix these, and sometimes they just say “we checked empirically” when it should be a theoretical argument.

**Reviewer Scores:**

The reviewers who were positive but still cautious probably won’t change much, maybe keep the same score or go a bit lower after discussion, since the big technical doubts weren’t really fixed. The critical ones, the ones who said there are proof bugs and big flaws, will almost certainly stick with reject or strong reject because their main objections are still there.

---

### Decision · Program_Chairs · 2026-01-26

Reject